# SeqPATE: Differentially Private
# Text Generation via Knowledge Distillation

**Zhiliang Tian**[1]*, **Yingxiu Zhao**[2], **Ziyue Huang**[2], **Yu-Xiang Wang** [3], **Nevin L. Zhang** [2], **He He** [4]
[1] National University of Defense Technology,
[2] The Hong Kong University of Science and Technology,
[3] UC Santa Barbara,
[4] New York University
tianzhilianghit@gmail.com, yzhaocx@connect.ust.hk, zyhuang94@gmail.com,
yuxiangw@cs.ucsb.edu, lzhang@cse.ust.hk, hhe@nyu.edu

## Abstract

Protecting the privacy of user data is crucial for text generation models, which can leak sensitive information during generation. Differentially private (DP) learning methods provide guarantees against identifying the existence of a training sample from model outputs. PATE is a recent DP learning algorithm that achieves high utility with strong privacy protection on training samples. However, text generation models output tokens sequentially in a large output space; the classic PATE algorithm is not customized for this setting. Furthermore, PATE works well to protect sample-level privacy, but is not designed to protect phrases in samples. In this paper, we propose SeqPATE, an extension of PATE to text generation that protects the privacy of individual training samples and sensitive phrases in training data. To adapt PATE to text generation, we generate pseudo-contexts and reduce the sequence generation problem to a next-word prediction problem. To handle the large output space, we propose a candidate filtering strategy to dynamically reduce the output space, and refine the teacher aggregation of PATE to avoid low agreement due to voting for a large number of candidates. To further reduce privacy losses, we use knowledge distillation to reduce the number of teacher queries. The experiments verify the effectiveness of SeqPATE in protecting both training samples and sensitive phrases.

## 1 Introduction

Recent work has shown that sensitive user information in training corpora, such as addresses and names, can be extracted from text generation models [6]. Providing privacy guarantees to the training corpora of text generation models has become a critical problem. Differential privacy (DP) provides provable guarantees against detecting individuals in datasets. Deep learning models with DP guarantees ensure that the existence of a specific training sample cannot be detected.

NoisySGD [42, 3, 1] is a popular DP algorithm for deep learning that adds noise to the gradients. PATE [31] is another type of DP learning algorithm that transfers knowledge from teachers trained on private data to a student model, where noises are added to teacher predictions to satisfy DP. PATE is model-agnostic, and its privacy cost derives from the knowledge distillation process instead of the model gradients in NoisySGD [42, 24]. Therefore, the noises required by PATE do not scale with model size. Given this benefit, PATE has great potential for text generation, since large language

---

*This paper was partially done when Zhiliang Tian was a Ph.D. student at HKUST and a visiting scholar at NYU.

36th Conference on Neural Information Processing Systems (NeurIPS 2022).

models (e.g., GPT-2 [35]) have become the backbone of most text generation models. However, NoisySGD and PATE are used to protect sample-level privacy [51, 24] and not customized to protect sensitive phrases in the data with a low privacy cost [22, 39, 50]. Additionally, PATE, originally designed for classification tasks, is not customized for sequential generation on a large output space (i.e., the natural language vocabulary), which is very common in text generation.

In this paper, we propose SeqPATE, a DP learning algorithm for text generation to protect the privacy of training corpora. By satisfying DP, SeqPATE has the guarantee of preventing the existence of training samples and sensitive phrases in the training corpora from being detected. Similarly to PATE, SeqPATE employs a teacher-student framework: (i) a student model learns to generate text from non-sensitive samples; and (ii) a number of teacher models, trained on sensitive text, supervise the student through noised outputs of aggregated teachers. The calibrated noise added to the output ensures that SeqPATE satisfies the DP requirements. This framework still faces some challenges in text generation. First, it suffers from the high costs of GPU memory and time. To obtain sentence-level supervision for text generation, the model needs to roll out all teachers to produce a sentence (i.e. all teachers vote to generate a word, which is then used as the input for the next word prediction). It results in a high inference cost with a large number of teachers (e.g. $2k$ teachers which are common in PATE). Second, the large output space (i.e., the vocabulary) in text generation leads to (i) low agreement rates among teachers and (ii) large noises required by DP, both of which significantly hurt the task performance.

To address the challenges, we generate pseudo-data using a pre-trained language model so that teachers only need to provide token-level supervision given the pseudo inputs. To handle the large output space and reduce the noise, we propose to dynamically filter the candidate words and select only words with high probabilities. Also, we aggregate teachers' outputs by interpolating their output distributions instead of voting with argmax predictions. DP learning methods provide privacy protection by adding noise, which also reduces the utility of the model. To reduce utility loss, we avoid unnecessary knowledge distillation by selectively applying knowledge distillation to generation steps where the student struggles. Most DP learning methods, including SeqPATE, prevent samples from being extracted. SeqPATE has further advantages in protecting users' secret phrases that occur multiple times in the corpora. We evaluate SeqPATE on a sentence completion task, which demonstrates its advantage in protecting samples and phrases compared to the baselines.

Our contribution is twofold: (i) We propose SeqPATE that provides privacy at both the sample level and the phrase level with theoretical analyses. (ii) We propose several strategies for SeqPATE to handle autoregressive text generation models with a large vocabulary.

## 2 Problem Setup

Our goal is to achieve the privacy protection quantified by DP in text generation to prevent attackers from inferring whether a sample or an n-gram appears in the training set. Our setting contains two types of textual datasets: (1) a private set $\mathcal{D}^{\text{pri}}$ from a corpus with sensitive information, (2) a public set $\mathcal{D}^{\text{pub}}$ that contains no sensitive information or comes from data contributors (e.g., volunteers) who have no objection to publishing their data. We aim to protect the privacy on the private set and can ignore the privacy protection on the public set.

Our application, sentence completion, aims to complete the whole sentence given the prefix. We train a language model to accomplish the task. The public set $\mathcal{D}^{\text{pub}}$ consists of prefixes, which can hardly contain sensitive information. The private set $\mathcal{D}^{\text{pri}}$ consists of whole sentences. Such a setting fits some real-world text generation applications: in dialog systems, the training samples from online services consist of questions and responses. The questions from customer service staff or service robots can be public, and the response from users carrying individual information should be private.

## 3 Background on DP and PATE

**Definition 3.1.** [Differential privacy (DP) [13, 14]] For any two neighboring datasets $\mathcal{D}, \mathcal{D}'$ (differ in only one individual), a randomized algorithm $\mathcal{M} : \mathcal{X}^n \to \mathcal{Y}$ is $(\varepsilon, \delta)$-differentially private if,

$$\Pr[\mathcal{M}(\mathcal{D}) \in S] \le e^{\varepsilon} \cdot \Pr[\mathcal{M}(\mathcal{D}') \in S] + \delta, \ \ \forall S \subseteq \mathcal{Y}, \ \text{where} \ \varepsilon > 0, \ \delta \ge 0. \tag{1}$$

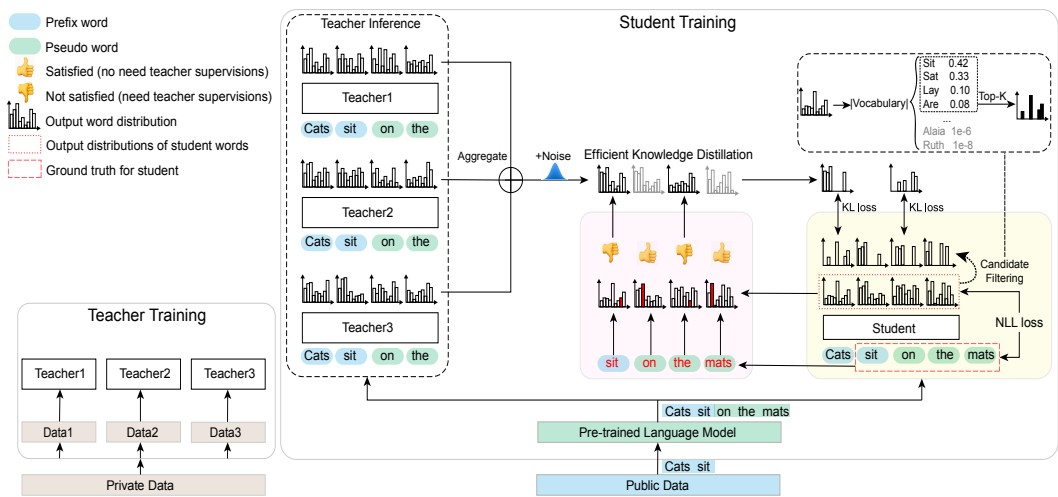

Figure 1: Overview of SeqPATE. SeqPATE trains teachers on private data. Student models are trained on pseudo-sentences generated by a pre-trained language model given the public prefixes. The student is supervised by aggregated teacher output distributions. SeqPATE benefits from candidate filtering (white block in the top right corner) and efficient knowledge distillation that determines whether teacher supervision is needed (pink block).

By definition, DP is a quantifiable definition of privacy that provides guarantees on identifications of individual data (preventing an adversary from inferring whether the input is $\mathcal{D}$ or $\mathcal{D}'$). ML models with DP ensure that each training sample has a degree of *plausible deniability*, i.e., the trained model is *just as likely as* to be trained on an alternative dataset *without* that sample. In SeqPATE, $\mathcal{M}$ is the entire training and inference process, $S$ is the vocabulary, and $Pr[\cdot]$ denotes the output distribution of generating a word. Attackers cannot tell whether a sample is in the training set or not, since the output distributions of the datasets *with or without that sample* are very similar (bounded by Eq. 1).

PATE [31], designed for classification tasks, takes advantage of an unlabeled public dataset $\mathcal{D}^{\text{pub}}$ and also trains on a labeled private set $\mathcal{D}^{\text{pri}}$ in a semi-supervised scenario. PATE achieves DP through a teacher-student framework with $M$ teacher models and a student model, where the student learns from the private set via knowledge distillation through teachers. PATE has three parts: (i) **The teacher models** are trained on the private set $\mathcal{D}^{\text{pri}}$, which is shuffled and divided into $M$ disjoint subsets. Each teacher is trained on one subset. (ii) **Teacher aggregation** merges the teachers' outputs. Each of the trained teachers then provides supervision to the student's unlabeled public set $\mathcal{D}^{\text{pub}}$. We use noised majority votes from teachers as labels to supervise the student. (iii) **A student model** is trained on the public set $\mathcal{D}^{\text{pub}}$ with the supervision of the aggregated teachers.

## 4  Approach

Fig. 1 shows an overview of SeqPATE. Given the public prefix (e.g., "Cats sit"), we first obtain the pseudo-inputs by completing the sentence (e.g., "Cats sit on the mats") using a pre-trained language model (Sec. 4.1). At each word, we then aggregate the teachers' prediction of the next word as supervision for training the student model (Sec. 4.2). To reduce the noise required by DP for a large output space of the size of the vocabulary, we reduce the output space by dynamically filtering unimportant words. To reduce the number of teacher queries that incur privacy losses, we propose an efficient knowledge distillation strategy that only queries teacher labels on uncertain examples (Sec. 4.3). We show the training algorithm in App. B and a running example in App. K.

### 4.1  Pseudo Input Generation

Conventional text generation models generate words sequentially from left to right. Thus, naively applying PATE to text generation requires rolling out all teachers word by word, i.e., iteratively sampling the next word from the aggregated teacher prediction. This is costly in both computation (running inference for hundreds of teacher models) and privacy costs (querying teachers at every step).

To tackle this challenge, we use a pre-trained language model to complete the public prefixes into pseudo sentences; thus, we only need to query teachers on the next word given a (pseudo) context.

## 4.2 Teacher Aggregation

PATE aggregates teacher predictions by majority vote. While it works for classification problems with a relatively small number of classes, the output space of text generation models contains all words in the vocabulary. As a result, the number of votes for each candidate word may be very low without a clear winner. For example, multiple candidates may tie for the top-1 prediction.

Inspired by Chen et al. [9, 17], we aggregate teacher results by averaging their output distributions. We first train $M$ teacher models on disjoint subsets of the private data. To produce the aggregated next word distribution given a context $c$, we average the teachers' output distributions, add calibrated noises, and then renormalize the results into a proper distribution. Following Papernot et al. [32], we apply the Gaussian mechanism. Formally, let $p_\phi^m(\cdot \mid c)$ be the prediction of the $m$-th teacher. The aggregated distribution is $p_{\text{agg}}(\cdot \mid c) \propto \frac{1}{M} \sum_{m=1}^{M} (p_\phi^m(\cdot \mid c) + \mathcal{N}(0, \sigma^2))$, [2] where the Gaussian noise is added to the aggregated output distribution. The way of SeqPATE satisfies DP guarantee (Eq. 1) is to add that calibrated noise to the teachers' output as mentioned above (detailed analyses in Sec. 5).

## 4.3 Training of the Student Model

The student model is trained on public pseudo-data and also supervised by the aggregated teachers.

**Training objectives.** The student model is a language model that predicts the next word given prior contexts. Given contexts from the (public) pseudo-data autocompleted by a pre-trained language model (GPT-2), the student is supervised by both the aggregated teacher predictions and the next word in the pseudo-data (i.e. pseudo label). The pseudo-data acts as a prior for the student given that the number of teacher queries is limited due to privacy concerns. The student's loss function has two parts:

- $\mathcal{L}_{\text{teacher}}$ denotes the loss with respect to teacher supervision. Note that the aggregated teacher output is a distribution over words. Therefore, we minimize the forward KL divergence between the aggregated teacher distribution $p_{\text{agg}}$ and the student output distribution $p_\theta$:

$$\mathcal{L}_{\text{teacher}}(c, p_{\text{agg}}) = \text{KL}\left(p_{\text{agg}}(\cdot \mid c) \,\|\, p_\theta(\cdot \mid c)\right). \tag{2}$$

- $\mathcal{L}_{\text{pseudo}}$ denotes the loss with respect to the pseudo-labels $w$ from $\tilde{\mathcal{D}}^{\text{pub}}$ (i.e. next words generated by a generic language model). Similar to standard language modeling, we use the negative log-likelihood:

$$\mathcal{L}_{\text{pseudo}}(c, w) = -\log p_\theta(w \mid c). \tag{3}$$

Eq. 4 shows the complete loss. ($\lambda$ balances the two terms and we discuss the noise scale $\sigma$ in Sec. 5.)

$$\mathcal{L}(p_{\text{agg}}, \tilde{\mathcal{D}}^{\text{pub}}) = \sum_{(c,w) \in \tilde{\mathcal{D}}^{\text{pub}}} \mathcal{L}_{\text{pseudo}}(c, w) + \lambda \mathcal{L}_{\text{teacher}}(c, p_{\text{agg}}), \tag{4}$$

**Reducing the output space via candidate filtering.** The high-dimensionality of the output of text generation models results in large noise (which is added to each coordinate). To reduce the output dimension (hence the amount of noise), we filter words on the tail of the distribution of the student model (i.e. set their probability to zero), and renormalize the teacher's aggregated distribution and the student output distribution over the rest words.

Note that the candidate filtering is based on the student's outputs on public or already released inputs, thus it does not affect the privacy guarantee. This choice improves the privacy-utility tradeoff by adaptively allocating the privacy budget to release the information most helpful to the task.

We experiment with two filtering strategies: top-$k$ and top-$p$. In top-$k$ filtering, we retain only the top-$k$ most likely candidates and filter the rest according to the student model. In top-$p$ filtering [18],

---

[2]Mathematically, the aggregated distribution with noises may be negative. If so, we renormalize the negative value to 0. Practically, we observed that being negative is an extremely rare event, since the $M$ is usually very large (e.g., $2k$) and the first term dominates the above equation.

$k$ is chosen dynamically such that the top-$k$ words are the minimum set whose cumulative probability is at least $p$. The strategy seldom loses good candidates because the student usually does well on top-$k$ predictions since the beginning of the training. [3]

**Reducing the number of teacher queries via efficient knowledge distillation.** While the aggregated teacher model satisfies DP, each query from the student incurs some privacy loss. Therefore, we obtain teacher supervision only on "hard" examples when training the student. Note that the student is trained on both the pseudo-data and local supervision from the teachers. We consider an example to be hard if the student cannot imitate the pseudo-label, in which case distilling knowledge from the teachers that are trained on large private data is helpful.

Concretely, we query teachers only when the rank of the pseudo-label is below a certain threshold among words ordered by descending probabilities under the student model. If we query the teachers, the student is trained via complete loss $\mathcal{L}(p_{\text{agg}}, \tilde{\mathcal{D}}^{\text{pub}})$ (Eq. 4); otherwise, the student is trained via the $\mathcal{L}_{\text{pseudo}}$ (Eq. 3). We note that the selection of tokens relies only on the student and is independent of the teachers; thus, the selection does not cause any additional privacy loss.

## 5 Privacy Analyses

### 5.1 Preliminary of Differential Privacy

**Lemma 5.1** (Analytical Gaussian mechanism [2]). *For a numeric query $f : \mathcal{X}^n \rightarrow \mathbb{R}^d$ over a dataset $\mathcal{D}$, the randomized algorithm that outputs $f(\mathcal{D}) + Z$ where $Z \sim \mathcal{N}(0, \sigma^2 I_d)$ satisfies $(\varepsilon, \delta(\varepsilon))$-DP for all $\varepsilon \geq 0$ and $\delta(\varepsilon) = \Phi(\frac{\Delta}{2\sigma} - \frac{\varepsilon\sigma}{\Delta}) - e^\varepsilon \Phi(-\frac{\Delta}{2\sigma} - \frac{\varepsilon\sigma}{\Delta})$. where $\Delta := \Delta_2^{(f)} = \max_{\mathcal{D} \sim \mathcal{D}'} \|f(\mathcal{D}) - f(\mathcal{D}')\|_2$ is the global L2 sensitivity of $f$ and $\Phi$ is the CDF function of $\mathcal{N}(0, 1)$.*

We can use the same result for an adaptive composition of a sequence of Gaussian mechanisms.

**Lemma 5.2** (Composition of Gaussian mechanisms [11]). *The adaptive composition of a sequence of Gaussian mechanisms with a noise level $\sigma_1, \sigma_2, \ldots$ and global L2 sensitivity $\Delta_1, \Delta_2, \ldots$ satisfies $(\varepsilon, \delta(\varepsilon))$-DP for all $\varepsilon \geq 0$ and $\delta(\varepsilon) \leq \delta_{\mathcal{M}}(\varepsilon)$ where $\mathcal{M}$ is a Gaussian mechanism with noise multiplier $\sigma/\Delta = \left(\sum_i (\Delta_i/\sigma_i)^2\right)^{-1/2}$.*

Specifically, the adaptive composition of a $k$ identical Gaussian mechanism with a noise multiplier $\sigma$ satisfies the same privacy guarantee of a single Gaussian mechanism with a noise multiplier $\sigma/\sqrt{k}$. By fixing $k$ and $\varepsilon$, we can calibrate the noise by choosing an appropriate $\sigma$ in Sec. 4.2.

### 5.2 Differential Privacy for Language Models at the Sample Level

Recall that we partition the private dataset into $M$ disjoint subsets, and train each teacher model on one of the subsets. Let vector $x_i \in \mathbb{R}^{|\mathcal{V}|}$ denote the probability distribution predicted by the $i$-th teacher model given some context, where $|\mathcal{V}|$ is the vocabulary size. The aggregation function $f(\mathcal{D}) := \sum_{i=1}^M x_i$ is the sum of the probability distributions predicted by all teachers. Since the datasets are disjoint, changing one sample affects only one teacher model. For neighboring datasets $\mathcal{D}, \mathcal{D}'$, let $j$ denote the index of each teacher model; the probability distributions $x_j$ and $x_j'$ (derived from $\mathcal{D}$ and $\mathcal{D}'$ respectively) are different. Then, the sensitivity $\Delta$ in Lemma 5.1 & 5.2 is (See detailed deductions in App. C),

$$\Delta := \Delta_2^{(f)} = \|f(\mathcal{D}) - f(\mathcal{D}')\|_2 \leq \|x_j - x_j'\|_2 \leq \sqrt{2}.$$

Adding the noises given by Lemma 5.2 to each coordinate (each candidate at each generation step of SeqPATE) preserves $(\varepsilon, \delta(\varepsilon))$-DP for $f(\mathcal{D})$. Finally, when we extract top-$k$ coordinates by top-$k$ candidate filtering (Sec. 4.3), the privacy guarantee also holds due to the post-processing property [14]. Therefore, the fact about whether a sample is in SeqPATE's private sets is protected (satisfying $(\varepsilon, \delta(\varepsilon))$-DP).

---

[3] In the first 10 training batches, the top-50 predictions of the student cover 94% "true" labels of pseudo samples.

## 5.3 Differential Privacy of Users' Secret Phrases

The above analyses show that we can protect the privacy of each sample (i.e., one *occurrence* of a sentence). However, in practice, we may want to protect all occurrences of some *secret phrases* specific to a user (e.g., names and addresses).[4] Consider a secret phrase $s$ that occurs $n_s$ times ($n_s \geq 1$) in the private set. According to group privacy [14], the protection on phrase $s$ satisfies $(n\varepsilon, \frac{e^{n\varepsilon}-1}{e^{\varepsilon}-1}\delta)$-DP [22], where the privacy loss scales linearly with the number of occurrences of $s$ (We discuss and analyze a better strategy to reduce the privacy loss of baselines in App. M).

Naively applying a DP algorithm requires larger noise to protect phrases that may occur multiple times. SeqPATE enjoys a stronger guarantee by assigning all data of a single user to one or a few teachers, such that any user-specific phrase occurs in the training data of only one or a few teachers. We denote $\tilde{n}_s$ as the number of teachers whose data contain the phrase $s$. Since adding or removing the phrase $s$ affects only $\tilde{n}_s$ teachers ($\tilde{n}_s$ is usually 1 or 2) and thus results in a sensitivity of $\sqrt{2}\tilde{n}_s$ (See App. D for details). In this way, the strength of protection on secret phrases is roughly equal to that we have derived for sample-level DP. The exact $(\varepsilon, \delta(\varepsilon, \tilde{n}_s))$-DP for the phrase $s$ can be obtained according to Lemma 5.1 & 5.2, where $\delta(\varepsilon, \tilde{n}_s) = \Phi(\frac{\tilde{n}_s}{\sqrt{2}\sigma} - \frac{\varepsilon\sigma}{\sqrt{2}\tilde{n}_s}) - e^{\varepsilon}\Phi(-\frac{\tilde{n}_s}{\sqrt{2}\sigma} - \frac{\varepsilon\sigma}{\sqrt{2}\tilde{n}_s})$. Unlike other generic DP algorithms such as NoisySGD, SeqPATE avoids a linear increase in privacy loss (i.e., a linear increase in $\varepsilon$) on user phrases by careful partitioning of the private data.

This effect is complimentary to other generic, but more intrusive, techniques such as *redaction* and *deduplication* [50] for addressing the same issue. Finally, a user-specific partitioning with SeqPATE also protects multiple secret phrases of the same user (e.g., a combination of SSN, credit card numbers, address, day of birth) *jointly* without incurring a larger privacy loss — a benefit that deduplication does not provide.

## 5.4 How does DP prevent memorization in SeqPATE?

In practice, the privacy of the language model is usually interpreted as not generating a secret phrase in the training data *as-is* during inference. Thus, one may wonder how DP prevents such unintended memorization of the training data. We remark that the protection against memorization follows the definition of DP. Consider the attack by Carlini et al. [6], which uses a language model to predict a secret phrase $s$ given a prefix. By the closure to post-processing [14], the prediction also satisfies DP. We denote $\mathcal{W}$ as the undesirable event where SeqPATE generates the phrase $s$ verbatim. The DP definition implies that the probability of $\mathcal{W}$ to happen when $s$ is in the SeqPATE's private sets is at most $e^{\varepsilon}$ larger than the probability of an alternative SeqPATE model trained without $s$ in those sets. The chances for the latter model to generate text with $s$ are astronomically small. Hence, DP implies that the probability of $\mathcal{W}$ under the former model (i.e. any SeqPATE model in general) is small.

# 6 Experiments

## 6.1 Experimental Settings

**Datasets.** We evaluate our model on two datasets. AirDialog [47] consists of 1M utterances from customer service dialog on flight booking; Europarl_v6 consists of 2M English sentences collected from European Parliament.[5] (See details about datasets in App. E.)

**Baselines.** We compare SeqPATE with two DP baselines: (1) standard **NoisySGD** trained on the private data with calibrated noise on clipped gradients [1, 22] and further trained on public set $\mathcal{D}^{\text{pub}}$ without protection; (2) based on **NoisySGD**, **NoisySGD+GC** [24] applies a ghost clipping which enables large batch size with memory saving techniques.

Additionally, we use two non-DP methods as reference: (1) **Pri-GPT** trained on the private set without any privacy protection; (2) the public pre-trained GPT-2 model **Pub-GPT** without access to private data. For all methods, we can optionally fine-tune on the generated pseudo-data as a warm-up, and the operation is denoted as $+\tilde{\mathcal{D}}^{\text{pub}}$.

---

[4]A formal definition of this is called personalized differential privacy, first seen in [16].

[5]www.statmt.org/europarl

**Implementation details.** All models are fine-tuned from the (public) pre-trained GPT-2 model [35]. The batch size is 32 for all comparing methods except the GC [24] (GC [24] requires 2048). We use Adam [23] and adjust the initial learning rate with a range of $10^{-3}$ to $10^{-6}$ for all methods. The $\delta$ mentioned in Sec. 5 for all DP methods is $10^{-6}$.

For SeqPATE, before training the student model with teacher supervision, we first fine-tune it on the public pseudo-data $\tilde{\mathcal{D}}^{\mathrm{pub}}$ as a warm-up. The coefficient $\lambda$ that balances supervision for the teacher and the pseudo-data (Eq. 4) is set to 20, where we have tuned it on the validation set of the public pseudo-data. The default number of teacher models is $2k$, where our model works well according to the experiments in App. H. We designed some strategies [6] to reduce memory and disk usage (See strategies and the computational cost in App. I). We run SeqPATE with $2k$ teachers on a single GPU in 3 days. Our code is publicly accessible. [7]. (See details about hyperparameters in App. G.)

**Evaluation Metrics.** We evaluate the generated text by perplexity (PPL) and Bleu (Bleu-n) [33].

## 6.2 Overall Performance

Table 1: The performance on the two datasets with sample-level protections (mentioned in Sec. 5.2). All SeqPATE results are statistically significant compared to the strongest baseline under paired sample t-test ($p < 0.05$).

| | | AirDialog | | | Europarl_v6 | | |
|---|---|---|---|---|---|---|---|
| | | PPL $\downarrow$ | Bleu-3 $\uparrow$ | Bleu-4 $\uparrow$ | PPL $\downarrow$ | Bleu-3 $\uparrow$ | Bleu-4 $\uparrow$ |
| | Pri-GPT | 3.88 | 21.51 | 17.16 | 23.25 | 1.77 | 0.86 |
| Non-DP | Pub-GPT | 63.16 | 0.31 | 0.10 | 57.40 | 1.02 | 0.35 |
| | Pub-GPT+$\tilde{\mathcal{D}}^{\mathrm{pub}}$ | 19.39 | 0.71 | 0.25 | 45.40 | 1.38 | 0.52 |
| | NoisySGD | 17.49 | 1.97 | 0.96 | 37.31 | 1.28 | 0.46 |
| DP (sample) | NoisySGD+$\tilde{\mathcal{D}}^{\mathrm{pub}}$ | 16.78 | 2.21 | 1.09 | 37.69 | 1.31 | 0.42 |
| $\varepsilon = 3$ | NoisySGD+GC+$\tilde{\mathcal{D}}^{\mathrm{pub}}$ | 11.17 | 3.15 | 1.54 | 35.77 | 1.56 | 0.57 |
| | SeqPATE | **8.00** | **5.09** | **3.24** | **33.92** | **1.60** | **0.61** |

**Protection at the sample level.** Tab. 1 show the performance on the two datasets. Among the non-DP baselines, Pri-GPT acts as an upper bound on the performance, since it can fully utilize the private set by discarding privacy protection. Pub-GPT+$\tilde{\mathcal{D}}^{\mathrm{pub}}$ outperforms Pub-GPT on both datasets, showing that the pseudo data is helpful (additional ablation study on the pseudo data in App. J also verifies this). NoisySGD+GC+$\tilde{\mathcal{D}}^{\mathrm{pub}}$ surpasses the above two methods, since it uses a much larger batch size (2048 vs 32) than NoisySGD. Our method, SeqPATE, significantly outperforms NoisySGD+GC+$\tilde{\mathcal{D}}^{\mathrm{pub}}$ (+59% in Bleu4 on AirDialog and +7.0% in Bleu4 on Europarl_v6) while ensuring the same level of privacy protection in terms of $\varepsilon$.

**Protection on the user's secret phrases.** We evaluate our method for privacy protection of secret phrases mentioned in Sec 5.3. The key step is to partition the data such that each phrase only occurs in the training data of very few teachers, which is straightforward given the user ID associated with the private data. In general, SeqPATE works with any set of secret phrases. In our experiments, we consider a user's full name as their secret phrase since it can be easily recognized from the data. We partition AirDialog's private data according to the accompanying user IDs. As a result, there are 96.6% users whose data are assigned to a single teacher (details about the data partition in App. F).

As described in Sec. 5.3, standard DP methods incur larger privacy loss on secret phrases. In Tab. 2, we see that NoisySGD+GC+$\tilde{\mathcal{D}}^{\mathrm{pub}}$ needs large noise to achieve a satisfactory level of protection on phrases, because $\varepsilon$ increases linearly with the frequency of the phrase (group privacy [14]). "Batching users" indicates partitioning data into batches according to users, which helps NoisySGD protect users' phrases (more analyses in App. M). For SeqPATE, the number of teachers trained on data containing the phrase $\tilde{n}_s$ is close to 1 on average after our partition. Thus, SeqPATE provides the same level of protection on users' secret phrases with a smaller noise and thus achieves better performance (+70% and +36% in Bleu4) (see more about the protection level on users' secret phrases in App. F).

---

[6]We train and conduct the inference on the teachers one-by-one and cache the teachers' outputs.

[7]https://github.com/tianzhiliang/SeqPATE

Table 2: The performance on AirDialog with the protections of users' secret phrases (mentioned in Sec. 5.3). $\varepsilon_{\text{avg}}$ is the average $\varepsilon$ over all secret phrases, as $\varepsilon$ of each phrase varies with the frequency of the phrase and the number of teachers (see App. F for detailed analyses about $\varepsilon_{\text{avg}}$). All results of SeqPATE are statistically significant compared to the strongest baseline under paired sample t-test ($p < 0.05$).

| | | PPL $\downarrow$ | Bleu-3 $\uparrow$ | Bleu-4 $\uparrow$ |
|---|---|---|---|---|
| DP (phrase) $\varepsilon_{\text{avg}} = 3$ | NoisySGD+GC+$\tilde{\mathcal{D}}^{\text{pub}}$ | 16.75 | 1.71 | 0.57 |
| | NoisySGD+GC+$\tilde{\mathcal{D}}^{\text{pub}}$ (batching users) | 13.42 | 3.25 | 1.45 |
| | SeqPATE | **10.10** | **4.20** | **2.46** |
| DP (phrase) $\varepsilon_{\text{avg}} = 5$ | NoisySGD+GC+$\tilde{\mathcal{D}}^{\text{pub}}$ | 16.49 | 1.89 | 0.69 |
| | NoisySGD+GC+$\tilde{\mathcal{D}}^{\text{pub}}$ (batching users) | 10.56 | 4.60 | 2.87 |
| | SeqPATE | **8.06** | **6.10** | **3.90** |

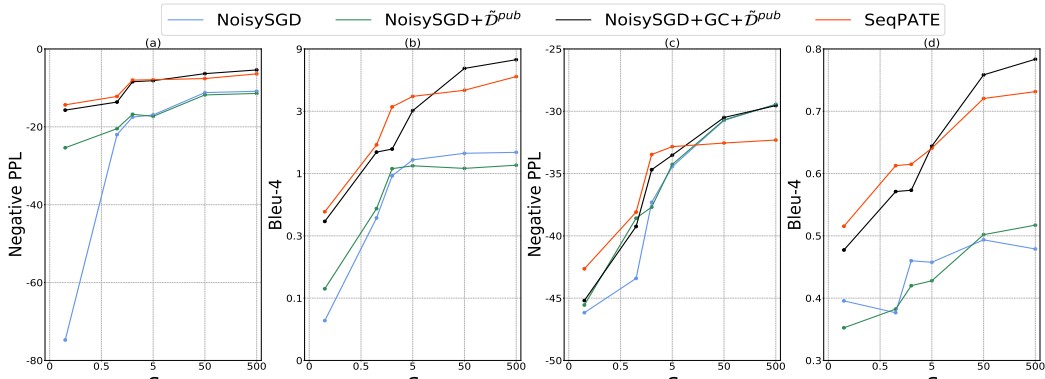

Figure 2: The private-utility tradeoff in Bleu-4 and PPL on a different $\varepsilon$. All the results are under sample level protections. Subfigure a & b show the results on AirDialog; c & d show the results on Europarl_v6. The grey lines show the "lower bound" since the method does not access the private set.

**Privacy-utility tradeoff.** In Fig. 2, we show the private-utility tradeoff curve of all DP algorithms.[8] Typically, DP with $\varepsilon \in [0.1, 10]$ is considered to provide a meaningful protection [45]. We observe that SeqPATE outperforms NoisySGD and NoisySGD+GC+$\tilde{\mathcal{D}}^{\text{pub}}$ in this range. However, SeqPATE does not work better than the two methods when $\varepsilon > 10$. The reason is that NoisySGD+GC+$\tilde{\mathcal{D}}^{\text{pub}}$ approaches Pri-GPT as $\varepsilon$ approaches infinity (i.e. the noise approaches 0). However, SeqPATE with an infinite $\varepsilon$ is still weaker than Pri-GPT because distillation still incurs performance loss: the teachers cannot completely transfer knowledge from the private data to the student. Therefore, we suggest using SeqPATE if strong privacy protection is desirable.

Table 3: Ablation studies. "$-$" means not using that strategy.

| | AirDialog | | | Europarl_v6 | | |
|---|---|---|---|---|---|---|
| | PPL $\downarrow$ | Bleu-3 $\uparrow$ | Bleu-4 $\uparrow$ | PPL $\downarrow$ | Bleu-3 $\uparrow$ | Bleu-4 $\uparrow$ |
| SeqPATE | 8.00 | 5.09 | 3.24 | 33.92 | 1.60 | 0.61 |
| $-$Merge_P | 11.96 | 3.14 | 1.85 | 39.19 | 1.40 | 0.47 |
| $-$KL | 12.08 | 3.26 | 1.81 | 39.81 | 1.41 | 0.52 |
| $-\mathcal{L}_{\text{pseudo}}$ | 8.11 | 4.74 | 3.17 | 33.81 | 1.58 | 0.60 |
| $-$Effi KD | 9.37 | 4.45 | 3.02 | 34.10 | 1.57 | 0.57 |
| $-$Gaussian | 9.54 | 4.33 | 2.78 | 35.31 | 1.54 | 0.55 |
| $-$All | 13.21 | 2.95 | 1.69 | 42.74 | 1.32 | 0.44 |

---

[8]For the models without protections, we consider $\varepsilon$ to be zero for baselines using the public data and $\varepsilon$ to be infinity for baselines using the private data.

## 6.3 Ablation Studies

There are several design choices in SeqPATE and we study the importance of each of them. In Tab. 3, we consider the following variants of SeqPATE: (1) $-$Merge_P: aggregating the teachers by voting instead of averaging their output distributions; (2) $-$KL: training the student using the cross-entropy loss with respect to teachers' top-1 prediction instead of KL divergence; (3) $-\mathcal{L}_{\text{pseudo}}$: not learning from the pseudo label (Eq. 3); (4) $-$Effi KD: querying teachers on all samples without selection; (5) $-$Gaussian: using the Laplace mechanism as the original PATE algorithm instead of the Gaussian mechanism; and (6) $-$All: using none of the above strategies, which is similar (although not equivalent) to the original PATE (the difference is that PATE needs to roll out all teachers (Sec. 4.1)).

Aggregating the teachers by voting and training with KL loss are the most important strategies for SeqPATE. The poor performance on $-$Merge_P shows that voting is not suitable for text generation. The reason is that voting over a large output space leads to low agreement rates. The results show that the $\mathcal{L}_{\text{pseudo}}$ loss makes little contribution to SeqPATE. The reason is that we have pre-trained on the student's training set via $\mathcal{L}_{\text{pseudo}}$ before the student's training. The promotion caused by efficient knowledge distillation (Effi KD) on AirDialog is larger than that on Europarl_v6, which shows that the "clever" student (e.g., models on AirDialog with low PPL and high Bleu) benefits more from this strategy. This is because the "clever" student can dramatically save the privacy cost and transfer it to where it would benefit the student most. The poor performance of $-$All verifies that the original PATE is not suitable for text generation.

Table 4: Analyses about the candidate filtering strategies.

| | AirDialog | | | Europarl_v6 | | |
|---|---|---|---|---|---|---|
| | PPL $\downarrow$ | Bleu-3 $\uparrow$ | Bleu-4 $\uparrow$ | PPL $\downarrow$ | Bleu-3 $\uparrow$ | Bleu-4 $\uparrow$ |
| top-$p$ | 8.00 | 5.09 | 3.24 | 33.92 | 1.60 | 0.61 |
| top-$k$=1 | 18.23 | 0.89 | 0.38 | 45.15 | 1.40 | 0.53 |
| top-$k$=10 | 12.47 | 3.47 | 1.95 | 35.94 | 1.55 | 0.54 |
| top-$k$=50 | 7.89 | 4.96 | 3.35 | 33.74 | 1.59 | 0.59 |
| top-$k$=100 | 8.78 | 4.64 | 3.17 | 34.48 | 1.60 | 0.62 |
| top-$k$=200 | 9.24 | 3.77 | 2.94 | 34.63 | 1.57 | 0.55 |

## 6.4 Analyses on Candidate Filtering and Teacher Numbers

To analyze candidate filtering with different filtering strategies, we conduct experiments on top-$p$ and top-$k$ filtering. As shown in Tab. 4, our full model employs the top-$p$ filtering (the threshold $p$ is 0.95) surpasses most variants with manually chosen $k$. Top-$k$ filtering ($k$ =50 or 100) also works well. Filtering with a too small $k$ ($k = 1$ or $k = 10$) implies discarding too much useful information from the supervision ($k = 1$ is different from $-$ KL in Tab. 3, which uses the Top-1 of teachers' results). Filtering with oversize $k$ results in unnecessarily large noises. Candidates with very small probabilities should be filtered during generation; however, random noises may increase their probabilities, so models may generate those words that are misled by the noise.

The results in App. H show that more teachers lead to better results when the number of teachers is in the range of $1 \sim 2k$. This is because the noise assigned to each teacher drops linearly as the number of teachers increases. Note that SeqPATE cannot always benefit from increasing the teacher numbers, because the scale of each teacher's data is linearly decreased as the teacher numbers go up. We choose $\varepsilon = 3$ on the sample level protection for all results in Tabs. 3, 4, and App. H.

Additionally, we conduct empirical comparisons and analyses of SeqPATE versus the original PATE in App. N. We show the effects of protections on users' secret phrases in App. O. We compare SeqPATE with another non-DP based baseline (i.e. blacklist based filtering) in App. P. We also conduct a case study in App. Q.

## 7 Related Work

Text generation models may leak user information through the generated texts [19, 7]. One direction of privacy protection is to protect author-level (user-level) information. The methods prevent attackers from inferring the author attributes (e.g., gender, age) [25] and the relationship between information

and authors [29]. Some researchers [40, 41] infer the membership (whether samples from a given author are used to train the model) given a black-box model. Some papers protect user privacy of training data against untrusted servers via federated learning [27, 10]. Another direction is to prevent attackers from extracting sensitive information in training sets by analyzing the outputs [30, 22], which is urgently needed [7]. Our SeqPATE focuses on this direction. In this direction, regularization methods [6, 43, 20] restrict the model capacity and prevent the model from memorizing exact training samples. Anonymization methods [26, 44] detect sensitive text and replace it with non-sensitive text. Unlike DP [14] methods, the above methods do not provide a quantifiable guarantee for privacy protection. Some researchers focus on protecting user privacy against untrusted servers via federated learning [27, 10].

Some researchers apply DP to text generation. For user-level privacy, ER-AE [4] augments the semantic information in the generated text to hide authors' writing styles from attackers. McMahan et al. [28] propose a recurrent language model with a DP guarantee against the identification of users. Note that the user-level privacy (relationships between users and their information) is different from the privacy of users' secret phrases in our model: Our model prevents individual user phrases from being detected. Some researchers apply NoisySGD to text generation to prevent sensitive training samples from being extracted: some of them [37, 39, 50] employ DP to protect a part of selected tokens; others [22, 49, 24] apply DP to protect both samples and all tokens, but the privacy cost on tokens is very high (Sec. 5.3). Our model falls into the latter category and reduces the privacy cost of tokens. Kerrigan et al. [22] apply NoisySGD [1] to text generation. Yu et al. [49] investigate fine-tuning strategies on pre-trained language models with NoisySGD. Li et al. [24] apply ghost clipping to pre-trained language models with NoisySGD and reduce memory usage. Shi et al. [38] apply DP to particular generation steps instead of training samples or n-grams. Brown et al. [5] analyze DP based method versus data sanitization of text generation models. Brown et al. [12] propose a efficient NoisySGD to speed up model training.

Differential privacy (DP) [13, 14] formally defines and quantifies privacy. ML models with DP guarantee [46, 15, 52] prevent the existence of individual training examples from being detected [6]. Some researchers protect the privacy of empirical risk minimization classifiers [8] and SVM [36] with DP. Following Song et al. [42], NoisySGD [1] achieves DP on deep learning models by adding noises to gradients. Pichapati et al. [34] adaptively clip the gradient in NoisySGD. PATE [31, 32] transfers the knowledge from teacher models trained on private sets with noises to a student model. KNN-PATE [51] refines PATE by accessing only the k-nearest neighbors from the private set. Jordon et al. [21] adversarially learn to generate synthetic data with discriminators trained by PATE. These methods are not customized for text generation models. Xie et al. [48] propose DPGAN to adversarially learn with a generator and a discriminator.

# 8    Conclusion

In this paper, we propose a novel framework, SeqPATE, to protect the privacy of the training data for text generation models with DP guarantees. SeqPATE achieves a good privacy-utility trade-off by leveraging both private and public data. As an extension of PATE, SeqPATE can handle the sequential generation paradigm with large output space at each step and is therefore adaptive to text generation models. We avoid rolling out teachers by providing pseudo-inputs for the teacher's inference and the student's training. We further reduce the output space by candidate filtering and limit privacy losses via efficient knowledge distillation. SeqPATE achieves a better performance with the sample-level protection and further provides much stronger protection on users' secret phrases. The limitations, ethical considerations, and social impacts of this paper are in App. A and L.

# 9    Acknowledgement

Research in this paper was supported by Hong Kong Research Grants Council under grand No. 16204920. HH is partly supported by the Samsung Advanced Institute of Technology (Next Generation Deep Learning: From Pattern Recognition to AI). YW is partially supported by NSF Award #2048091. The authors thank Mr. Wei Dong and Dr. Yiping Song for their help and insights on this paper.

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
