However, we should also consider some extreme situations where someone intentionally applies our model to illegal applications. Someone may employ text generation models to create fake news or misinformation and protect himself or herself from being detected. In this way, our model may be used for illegal applications, which is a kind of potential negative societal impact. In the future, we will add some constraints to our model so that our model cannot generate texts for illegal applications (e.g., fake news generation).

In addition, we know large $\varepsilon$ (e.g., $\varepsilon = 500$) cannot provide a meaningful protection. We should carefully use our model and cannot assume that the model is perfect no matter what parameters ($\varepsilon$ and $\delta$) we use. One possible unethical application is to collect the data from users who believe our model can fully protect their privacy. It means the users may ignore the strength of privacy protection (in terms of the value of $\varepsilon$ and $\delta$). That may result in a negative impact on society. Hence, we kindly remind the researchers, who will use this model, to pay more attention to the strength of privacy protection. Further, we should prevent some researchers from collecting data from users who do not have a correct understanding of our algorithm. We suggest that the researchers should ensure the users, who contribute their data, fully understand the risks in our model.

As for the two datasets in the experiments, the Europarl_v6 does not contain the personally identifiable information of the real user. The Airdialog dataset contains some personally identifiable information of users, which enables us to conduct experiments to verify the performance of privacy protection. Note that the dataset had been already published to the public. So, our work in this paper does not further release the user's personal information.

# B    Algorithm for the Training of SeqPATE

The pseudo code of SeqPATE's training procedure is shown in Algorithm 1.

# C    Detailed Deduction of the Sensitivity in Sample Level DP

We obtain the Equations in Sec. 5.2 of the paper body since $x_j$ and $x'_j$ are the probability distributions over the vocabulary $\mathcal{V}$.

$$\Delta_2^{(f)} = \|f(\mathcal{D}) - f(\mathcal{D}')\|_2 \leq \|x_j - x'_j\|_2 = \left( \sum_{v=1}^{|\mathcal{V}|} (x_{jv} - x'_{jv})^2 \right)^{1/2} \tag{5}$$

We know $(x_{jv} - x'_{jv})^2$ is smaller than $|x_{jv} - x'_{jv}|$ since $|x_{jv} - x'_{jv}| \in (0,1)$ for each $v$. Hence, we have,

$$\left( \sum_{v=1}^{|\mathcal{V}|} (x_{jv} - x'_{jv})^2 \right)^{1/2} \leq \left( \sum_{v=1}^{|\mathcal{V}|} |x_{jv} - x'_{jv}| \right)^{1/2} \leq \left( \sum_{v=1}^{|\mathcal{V}|} |x_{jv} + x'_{jv}| \right)^{1/2}$$

We know $|a + b| = a + b$ when $a, b \in (0,1)$, so we have,

$$\left( \sum_{v=1}^{|\mathcal{V}|} |x_{jv} + x'_{jv}| \right)^{1/2} = \left( \sum_{v=1}^{|\mathcal{V}|} x_{jv} + \sum_{v=1}^{|\mathcal{V}|} x'_{jv} \right)^{1/2} = \left( 1 + 1 \right)^{1/2} \leq \sqrt{2},$$

In summary, the upper bound of the sensitivity is,

$$\Delta_2^{(f)} = \|f(\mathcal{D}) - f(\mathcal{D}')\|_2 \leq \|x_j - x'_j\|_2 = \sqrt{2},$$

---

**Algorithm 1** Training procedure of SeqPATE

---

**Require:** $\mathcal{D}^{\text{pri}}, \mathcal{D}^{\text{pub}}$: datasets, $GPT$: a pre-trained GPT-2 model.
1: $\{f_\phi^m\}_{m=1}^M$: $M$ teacher models, $f_\theta$: a student model, $f_\Theta$: a student model for self pre-training,
2: $\{\phi^m\}_{m=1}^M \leftarrow GPT, \Theta \leftarrow GPT$ # Initialize teachers and the student for self pre-training.
3: $GPT$ generates a pseudo dataset $\tilde{\mathcal{D}}^{\text{pub}}$ based on $\mathcal{D}^{\text{pub}}$.
4: $\{\mathcal{D}_m^{\text{pri}}\}_{m=1}^M \leftarrow \mathcal{D}^{\text{pri}}$ # Divide private dataset into $m$ subsets.
5: **for all** $m$ in $M$ **do**
6:     Train teacher $f_\phi^m$ on $\mathcal{D}_m^{\text{pri}}$
7: **end for**
8: Teachers $\{\phi^m\}_{m=1}^M$ conduct inference on $\tilde{\mathcal{D}}^{\text{pub}}$ to get $p_\phi^m(\cdot \mid c)$ required in Sec. 4.2 for all samples.
9:
10: Train $f_\Theta$ on $\tilde{\mathcal{D}}^{\text{pub}}$ # self pre-training for the student.
11: $\theta \leftarrow \Theta$ # Initialize the student model.
12:
13: **while** not converge **do**
14:     **for all** batch of samples $\{S\}^{\text{batchsize}}$ in $\tilde{\mathcal{D}}^{\text{pub}}$ **do**
15:         Student $f_\theta$ conducts feed-forward on $\{S\}^{\text{batchsize}}$.
16:         **for all** sample $S$ in the batch $\{S\}^{\text{batchsize}}$ **do**
17:             **for all** token $w_i$ in sample $S$ **do**
18:                 $p_{\text{agg}}(\cdot \mid c) \propto \frac{1}{M} \sum_{m=1}^M (p_\phi^m(\cdot \mid c) + \mathcal{N}(0, \sigma^2))$ # Aggregate teachers' outputs
19:                 Select only top-$k$ or top-$p$ predicted tokens as student's output.
20:                 Obtain $\mathcal{L}_{\text{teacher}}$ and $\mathcal{L}_{\text{pseudo}}$ as Eq. 4 in the paper body. # Noise is added into $\mathcal{L}_{\text{teacher}}$ to protect the privacy.
21:                 Get $\mathcal{L}$ by combining $\mathcal{L}_{\text{teacher}}$ and $\mathcal{L}_{\text{pseudo}}$ # Knowledge distillation with active learning.
22:             **end for**
23:         **end for**
24:         Update $\phi$ respect to $\mathcal{L}$.
25:     **end for**
26: **end while**

---

# D  Detailed Deduction of the Sensitivity of the Privacy on Users' Secret Phrases

Here, we show the detailed deduction of obtaining the sensitivity of the privacy on users' secret phrases mentioned in Sec. 5.3 (in the paper body). we treat each user's secret phrase $s$ as a data point. As only $\tilde{n}_s$ teacher models can access the phrase $s$ in the private set, changing the phrase $s$ affects at most $\tilde{n}_s$ teacher models. We redefine the neighboring datasets $\mathcal{D}, \mathcal{D}'$ are two datasets differ at only one user's secret phrase $s$. It means the phrase $s$ occurs in one dataset but not in another one. Let $j$ denotes the index of each teacher model, and $\{x_1, ..., x_j, ..., x_a\}$ and $\{x'_1, ..., x'_j, ..., x'_b\}$ mean the teacher outputs affected by the phrase $s$ (for the datasets $\mathcal{D}$ and $\mathcal{D}'$). We have $a \leq \tilde{n}_s$ and $b \leq \tilde{n}_s$ since changing a secret phrase $s$ affects at most $\tilde{n}_s$ teacher models. We calculate the sensitivity of $f$ as follows.

$$\Delta_2^{(f)} = \|f(\mathcal{D}) - f(\mathcal{D}')\|_2 \leq \|\sum_{j=1}^a x_j - \sum_{j=1}^b x'_j\|_2 \leq \|\sum_{j=1}^{\tilde{n}_s} x_j - \sum_{j=1}^{\tilde{n}_s} x'_j\|_2$$

For the above equation, we obtain the following equation ($x_j$ and $x'_j$ are the probability distributions over the vocabulary $\mathcal{V}$).

$$\|\sum_{j=1}^{\tilde{n}_s} x_j - \sum_{j=1}^{\tilde{n}_s} x'_j\|_2 = \left(\sum_{v=1}^{|\mathcal{V}|}(\sum_{j=1}^{\tilde{n}_s} x_{jv} - \sum_{j=1}^{\tilde{n}_s} x'_{jv})^2\right)^{1/2} = \left(\sum_{v=1}^{|\mathcal{V}|}(\frac{\tilde{n}_s}{\tilde{n}_s}\sum_{j=1}^{\tilde{n}_s} x_{jv} - \frac{\tilde{n}_s}{\tilde{n}_s}\sum_{j=1}^{\tilde{n}_s} x'_{jv})^2\right)^{1/2}$$

$$= \left(\sum_{v=1}^{|\mathcal{V}|} \tilde{n}_s^2(\frac{\sum_{j=1}^{\tilde{n}_s} x_{jv}}{\tilde{n}_s} - \frac{\sum_{j=1}^{\tilde{n}_s} x'_{jv}}{\tilde{n}_s})^2\right)^{1/2} = \tilde{n}_s\left(\sum_{v=1}^{|\mathcal{V}|}(\frac{\sum_{j=1}^{\tilde{n}_s}(x_{jv} - x'_{jv})}{\tilde{n}_s})^2\right)^{1/2}$$

We know $\left(\frac{\sum_{j=1}^{\tilde{n}_s}(x_{jv}-x'_{jv})}{\tilde{n}_s}\right)^2 \in (0,1)$ since $|x_{jv}-x'_{jv}| \in (0,1)$. Then, we have $\left(\frac{\sum_{j=1}^{\tilde{n}_s}(x_{jv}-x'_{jv})}{\tilde{n}_s}\right)^2 \leq \left|\frac{\sum_{j=1}^{\tilde{n}_s}(x_{jv}-x'_{jv})}{\tilde{n}_s}\right|$. Hence, we have,

$$\tilde{n}_s\left(\sum_{v=1}^{|\mathcal{V}|}\left(\frac{\sum_{j=1}^{\tilde{n}_s}(x_{jv}-x'_{jv})}{\tilde{n}_s}\right)^2\right)^{1/2} \leq \tilde{n}_s\left(\sum_{v=1}^{|\mathcal{V}|}\left|\frac{\sum_{j=1}^{\tilde{n}_s}(x_{jv}-x'_{jv})}{\tilde{n}_s}\right|\right)^{1/2} = \tilde{n}_s\left(\frac{1}{\tilde{n}_s}\sum_{v=1}^{|\mathcal{V}|}\left|\sum_{j=1}^{\tilde{n}_s}(x_{jv}-x'_{jv})\right|\right)^{1/2}$$

$$\leq \tilde{n}_s\left(\frac{1}{\tilde{n}_s}\sum_{v=1}^{|\mathcal{V}|}\sum_{j=1}^{\tilde{n}_s}|x_{jv}-x'_{jv}|\right)^{1/2} = \tilde{n}_s\left(\frac{1}{\tilde{n}_s}\sum_{j=1}^{\tilde{n}_s}\sum_{v=1}^{|\mathcal{V}|}|x_{jv}-x'_{jv}|\right)^{1/2} \leq \tilde{n}_s\left(\frac{1}{\tilde{n}_s}\sum_{j=1}^{\tilde{n}_s}\sum_{v=1}^{|\mathcal{V}|}|x_{jv}+x'_{jv}|\right)^{1/2}$$

$$= \tilde{n}_s\left(\frac{1}{\tilde{n}_s}\sum_{j=1}^{\tilde{n}_s}\sum_{v=1}^{|\mathcal{V}|}(x_{jv}+x'_{jv})\right)^{1/2} = \tilde{n}_s\left(\frac{1}{\tilde{n}_s}\sum_{j=1}^{\tilde{n}_s}\left(\sum_{v=1}^{|\mathcal{V}|}x_{jv}+\sum_{v=1}^{|\mathcal{V}|}x'_{jv}\right)\right)^{1/2} = \tilde{n}_s\left(\frac{\sum_{j=1}^{\tilde{n}_s}(1+1)}{\tilde{n}_s}\right)^{1/2} = \sqrt{2}\tilde{n}_s$$

In summary, the upper bound of the sensitivity is,

$$\Delta_2^{(f)} = \|f(\mathcal{D}) - f(\mathcal{D}')\|_2 \leq \|\sum_{j=1}^{a}x_j - \sum_{j=1}^{b}x'_j\|_2 \leq \sqrt{2}\tilde{n}_s,$$

## E  Descriptions about the Datasets

The AirDialog dataset [48] [9] consists of 402,038 dialogues. Each dialogue consists of more than two utterances. We treat each utterance as a sample (i.e. sentence) in our sentence completion task. The Airdialog dataset contains some personally identifiable information about users. It contains the users' names of the dialog speakers, which enables us to conduct experiments to verify the performance of privacy protection. Note that the dataset had been already published to the public. In this way, We obtain the Europarl_v6 dataset from a machine translation benchmark [10], where we only use the monolingual English dataset with 2,015,440 raw sentences. The Europarl_v6 does not contain the personally identifiable information of the real user.

For the above datasets, we filter the short sentence with less than eight tokens. Then, the first four tokens act as the prefix, and the rest of the tokens act as the output (ground-truth). We split each datasets into a private set $\mathcal{D}^{\text{pri}}$ and a public set $\mathcal{D}^{\text{pub}}$. For the AirDialog dataset, the private set contains 0.95M/5K/50K samples for training/validation/testing, and the public set contains 40K/5K for training/validation. For the Europarl_v6 dataset, the private set contains 1.72M/10K/50K samples for training/validation/testing, and the public set contains 40K/5K for training/validation. The vocabulary size for the two datasets is set to 50K. We replace the tokens out of the vocabulary with a special token.

## F  Dataset Partitions and Experiments about the Protection on Users' Secret Phrases

To achieve the protection of users' secret phrases mentioned in Sec. 5.3, we partition the original dataset into teachers' training data with the following principles: (1) the teacher number for each user should be small; (2) the scales of the data for every teacher are roughly balanced.

There are 9131 users in AirDialog's private training set. As shown in Tab. 6, after the above processing, the number of users whose data are assigned to a single teacher is 8824; there are 263 users whose data occurs in 2 teachers' training data; there are 41 users whose data belong to 3 teachers; only 3 users' data are accessed by more than 3 teachers. On average, the number of teachers for each user is 1.039.

The users' secret phrases mentioned in Sec. 5.3 are often the phrases known by a few users (occurs in a few users' data). For a secret phrase $s$, the number of teachers accessing the phrase $\tilde{n}_s$ is very small,

---

[9]The dataset comes from https://github.com/google/airdialogue

[10]The description can be found at https://www.statmt.org/europarl. The data come from https://statmt.org/wmt11/training-monolingual.tgz

| # Users | 8824 | 263 | 41 | 3 |
|---|---|---|---|---|
| # Teachers for each user | 1 | 2 | 3 | > 3 |

Table 5: The statistical information of the teacher numbers for all users in the AirDialog's private training.

and therefore the protections provided by SeqPATE on those phrases are naturally strong (if many users know $s$, the $\tilde{n}_s$ is large and the protection is naturally weak). The secret phrases may be phone number, address, name, or SSN number. In the AirDialog dataset, the description of each sample contains the "user's full name". With that information, we can easily check the existence and count the frequency of the "user's full name" in each sample. So, we can easily evaluate the protections on secret phrases by treating "user's full name" as the secret phrase.

| Methods | $\varepsilon_{avg}$ | min $\varepsilon$ | max $\varepsilon$ | % $\varepsilon \leq$ average $\varepsilon$ |
|---|---|---|---|---|
| NoisySGD+GC+$\tilde{\mathcal{D}}^{pub}$ | 3 | 0.12 | 731.58 | 52.8% |
| SeqPATE | 3 | 2.85 | 25.64 | 95.4% |
| NoisySGD+GC+$\tilde{\mathcal{D}}^{pub}$ | 5 | 0.20 | 1219.31 | 52.8% |
| SeqPATE | 5 | 4.75 | 42.78 | 95.4% |

Table 6: The statistical information of $\varepsilon$ on all the users' secret phrases under the protection of different algorithms. The first three columns show the average/minimal/maximal $\varepsilon$ over all secret phrases. The last column indicates the percentage of secret phrases whose $\varepsilon$ is lower than the average $\varepsilon$.

In the experiments about protections on users' secret phrases, the $\varepsilon$ of each phrase is different. In NoisySGD+GC+$\tilde{\mathcal{D}}^{pub}$, the $\varepsilon$ of the phrase $s$ relies on the phrase frequency $n_s$; in SeqPATE, the $\varepsilon$ of the phrase $s$ relies on the teacher number $\tilde{n}_s$. Hence, the $\varepsilon_{avg}$ reported in "DP (phrase)" of the second Table in Sec. 6.2 means average $\varepsilon$ over all secret phrases. There are 6705 secret phrases in the AirDialog dataset. In the model training, we apply a same scale of noises to the algorithm and then calculate the exact $\varepsilon$ for all secret phrases. The rows 1 and 2 of Tab. 6 show the $\varepsilon$ of the models in rows 1 and 2 of the second Table in Sec. 6.2. If we fix the average $\varepsilon_{avg}$ at 3, the $\varepsilon$ of phrases on NoisySGD+GC+$\tilde{\mathcal{D}}^{pub}$ range from 0.12 to 731.58, and the $\varepsilon$ of phrases on SeqPATE range from 2.85 to 25.64. The rows 3 and 4 of Tab. 6 show the $\varepsilon$ of the models in rows 3 and 4 of the second Table in Sec. 6.2. If we fix the average $\varepsilon_{avg}$ at 5, the $\varepsilon$ of phrases on NoisySGD+GC+$\tilde{\mathcal{D}}^{pub}$ range from 0.20 to 1219, and the $\varepsilon$ of phrases on SeqPATE range from 4.75 to 42.74.

From the Tab. 6, we can observe that 95.4% phrases have a lower $\varepsilon$ (stronger protection) than the average $\varepsilon_{avg}$ (3 or 5). However, NoisySGD+GC+$\tilde{\mathcal{D}}^{pub}$ can only ensure 52.8% phrases enjoy a stronger protection than the average level. Hence, compared to NoisySGD+GC+$\tilde{\mathcal{D}}^{pub}$, SeqPATE can provide strong protection on more secret phrases, even if we assign the same $\varepsilon_{avg}$ to SeqPATE and NoisySGD+GC+$\tilde{\mathcal{D}}^{pub}$.

# G  Details about the Experimental Setting

All the comparing methods use the same base model, the GPT-small, which has 12 stacked layers as mentioned in the original paper [35]. The pre-trained GPT-2 model comes from the official website [11]. We truncate the sentences with the maximal sentence length of 40. In the top-$p$ strategy, the threshold $p$ is 0.95 (We have tried the threshold of $0.90 \sim 1.0$ and found $p = 0.95$ works well). The threshold in active learning mentioned in Sec. 4.3 is 10 or 5 (We also need to tune the parameter). The hyperparameter tuning is conducted on the validation set of the public pseudo data, so tuning does not introduce additional privacy losses. For all our experiments, we adopt *autodp* [46] — an open-source library that implements the analytical Gaussian mechanism for privacy accounting and calibration. We use the TESLA V100 GPU devices with 32GB memory on a Slurm HPC cluster.

---

[11]github.com/openai/gpt-2

## H Analyses about the Number of Teacher Models

Tabs. 7 & 8 show the analyses about SeqPATE's performance with different teacher numbers on our two datasets. We evaluate the performance with the sample level protection ($\varepsilon = 3$).

| | AirDialog | | | | |
|---|---|---|---|---|---|
| | PPL ↓ | B-1 ↑ | B-2 ↑ | B-3 ↑ | B-4 ↑ |
| #teacher=1 | 19.28 | 8.59 | 2.35 | 0.86 | 0.28 |
| #teacher=10 | 16.57 | 7.97 | 2.24 | 0.85 | 0.30 |
| #teacher=200 | 10.96 | 12.81 | 5.13 | 2.89 | 1.34 |
| #teacher=$2k$ | 8.00 | 15.14 | 8.30 | 5.09 | 3.24 |

Table 7: SeqPATE's performance with different teacher numbers on the AirDialog dataset.

| | Europarl_v6 | | | | |
|---|---|---|---|---|---|
| | PPL ↓ | B-1 ↑ | B-2 ↑ | B-3 ↑ | B-4 ↑ |
| #teacher=1 | 41.56 | 12.39 | 3.71 | 1.13 | 0.39 |
| #teacher=10 | 38.94 | 12.89 | 3.75 | 1.21 | 0.44 |
| #teacher=200 | 34.55 | 13.25 | 4.18 | 1.36 | 0.51 |
| #teacher=$2k$ | 33.92 | 13.75 | 4.69 | 1.60 | 0.61 |

Table 8: SeqPATE's performance with different teacher numbers on the Europarl_v6 dataset.

## I The Computational Cost of SeqPATE

It seems that our model requires huge computational resources and a costly infrastructure to run. However, our model can train and infer on a single GPU machine. In this section, we introduce some simple strategies we used in our implementation and also introduce the total computational cost of our model.

**Memory usage and hard-disk space usage.** Since our method uses a large number of teachers, the naive implementation of loading all teachers into the memory for aggregation is impractical. However, note that our algorithm only needs to access the teachers' top-$k$ predictions. Therefore, we train teacher models sequentially. Once a teacher model is trained, we obtain its top-$k$ predictions ($k$=200 at most in our experiments) on the public training data and save the results (i.e. $k$ probabilities). Then, we discard the teacher model. Finally, SeqPATE only needs the teacher's supervision on a small number of samples. In our experiments, training on 500∼1k teacher labeled samples is sufficient. Overall, saving teachers' inference results uses 8∼16GB. The memory usage is similar to that of a GPT2 model because we do not load all teacher models into the memory and instead run inference sequentially and merge teachers' predictions offline.

**Training time.** While we have a large number of teachers, each teacher is trained on only a small fraction of the entire dataset. Thus, the time it takes to train all teachers is roughly equal to the time of training a single GPT2 model on the full dataset (of 1∼2M samples in our experiments). In practice, the teachers' training time of SeqPATE on AirDialog dataset is roughly 1 or 2 days; their training time on Europarl_v6 dataset is 2 or 3 days. For both datasets, the student's training time is range from several minutes to half an hour. For the NoisySGD, the whole training takes 1 or 2 days. The running time of the inference for all methods is similar, which takes around 10 minutes.

In summary, with the simple strategies, the teacher training and aggregation steps are not much more expensive than training a GPT2 model. Compared to standard NLG model training, our algorithm does not require special hardware or distributed learning.

## J The Contribution of the Pseudo Public Dataset $\tilde{\mathcal{D}}^{\text{pub}}$

The following experiments verify the contribution of the pseudo-public dataset $\tilde{\mathcal{D}}^{\text{pub}}$ to our task. We conduct the experiments on the AirDialog dataset. The results in Tab. 9 shows that using

$\tilde{\mathcal{D}}^{\text{pub}}$ can prompt the performance a lot, where the promotion can be found in both SeqPATE and NoisySGD+GC+$\tilde{\mathcal{D}}^{\text{pub}}$ methods. SeqPATE relies highly on the pseudo data since the pseudo data provide the input text in the student's training.

| Methods | Dataset | PPL $\downarrow$ | B-1 $\uparrow$ | B-2 $\uparrow$ | B-3 $\uparrow$ | B-4 $\uparrow$ |
|---|---|---|---|---|---|---|
| NoisySGD+GC | w/ $\tilde{\mathcal{D}}^{\text{pub}}$ | 11.17 | 13.21 | 5.90 | 3.15 | 1.54 |
| NoisySGD+GC | w/o $\tilde{\mathcal{D}}^{\text{pub}}$ | 12.05 | 12.94 | 5.97 | 2.96 | 1.36 |
| SeqPATE | w/ $\tilde{\mathcal{D}}^{\text{pub}}$ | 8.00 | 15.14 | 8.30 | 5.09 | 3.24 |
| SeqPATE | w/o $\tilde{\mathcal{D}}^{\text{pub}}$ | 16.12 | 10.86 | 3.92 | 2.13 | 0.96 |

Table 9: The comparisons between using and not using the pseudo dataset, $\tilde{\mathcal{D}}^{\text{pub}}$.

# K    The Illustration of a Running Example

Here, we will use an example to show our training processing. In this example, the prefix from the public dataset $\mathcal{D}^{\text{pub}}$ is "I want to book". We feed the prefix to a pre-trained GPT-2 model to generate a pseudo sentence "I want to book a flight from Tokyo to Hawaii". The pseudo sentence serves as an example in the pseudo-public dataset $\tilde{\mathcal{D}}^{\text{pub}}$. We feed the pseudo sentence to the teacher models to conduct the teachers' inference; we also feed it to the student model to conduct the feed-forward of the student's training. Teacher models output the probability distributions on all words (10 words, in total) of the sentence. Then, we aggregate all teachers' probability distributions and add the calibrated noises to the aggregated distributions. The student model also generates the corresponding probability distributions on those words. We conduct the knowledge distillation with active learning and the top-$k$ or top-$p$ filtering over the student's probability distributions. For example, if the student model can do well on the words ("I", "want", "book", "flight", and "Tokyo"), the student queries the teachers' output distributions only on the rest of the words ("to", "from", "to", and "Hawaii"). The student is supervised by the teachers' outputs via the KL loss mentioned in Sec. 4.3. Besides, the student is always supervised by the $\mathcal{L}_{\text{pseudo}}$ loss on the whole pseudo sentence ("I want to book a flight from Tokyo to Hawaii"). Finally, the student model conducts back-propagation according to the above losses.

# L    Limitation of This Paper

Even if our proposed method obtains remarkable performance, this work still has some limitations. We will continue to focus on this topic and try to address those limitations in future work.

Firstly, compared to NoisySGD-based methods, our model is not good at handling "big $\varepsilon$" (the $\varepsilon$ is very large, e.g., $\varepsilon = 50$). As mentioned in Sec. 6.2, the phenomenon is reasonable since the knowledge distillation in our method cannot completely transfer knowledge but a very large $\varepsilon$ in NoisySGD results in a very small noise. Note that researchers usually treat that $\varepsilon$ ranging from 0.1 to 5 provides meaningful protections. Too large $\varepsilon$ can hardly provide sufficient protections for the data. $\varepsilon > 5$ says that an individual could be identified with the confidence of more than 99.33% [45]. Hence, this limitation is not a big issue for this paper.

Secondly, compared with other papers, our experiment processing may be complex since our default setting is to use $2k$ teacher models. It means that we need to train $2k$ teacher models to conduct the experiments. Fortunately, we design some strategies to enable all teacher models to train on a single GPU within 3 days (according to App. I), ); thus the computational cost is not very high. In practice, we can use a shell script to run the $2k$ teachers automatically so the operations in the experiment are not so heavy to conduct.

Thirdly, we have not applied our method to other text generation applications, such as machine translation and summarization. The privacy concern in those tasks is also an urgent need. We may try to apply SeqPATE to some new applications in the future. Since the state-of-the-art models in those applications may have some sophisticated components , we believe some further works are needed to apply our model to other text generation tasks.

## M    More Explanation of the Experiments on Users' Secret Phrases

SeqPATE achieves strong privacy protections on users' secret phrases by carefully partitioning the private set according to users. Note that NoisySGD (DP-SGD) can also follow the similar idea and partition the private set to training batches according to users, that is, we try to ensure a user's data is in one or a few batches. We call this method "batching users ". We conducted experiments in this way and reported the results in Table 13 (row 2 and 4). The experimental results show that our method on protecting users' secret phrases still outperforms the NoisySGD baselines with batching users.

| | | PPL ↓ | Bleu-3 ↑ | Bleu-4 ↑ |
|---|---|---|---|---|
| DP (phrase) $\varepsilon_{avg} = 3$ | NoisySGD+GC+$\tilde{\mathcal{D}}^{pub}$ | 16.75 | 1.71 | 0.57 |
| | NoisySGD+GC+$\tilde{\mathcal{D}}^{pub}$ (batching users) | 13.42 | 3.25 | 1.45 |
| | SeqPATE | **10.10** | **4.20** | **2.46** |
| DP (phrase) $\varepsilon_{avg} = 5$ | NoisySGD+GC+$\tilde{\mathcal{D}}^{pub}$ | 16.49 | 1.89 | 0.69 |
| | NoisySGD+GC+$\tilde{\mathcal{D}}^{pub}$ (batching users) | 10.56 | 4.60 | 2.87 |
| | SeqPATE | **8.06** | **6.10** | **3.90** |

Table 10: The performance on AirDialog with the protections of users' secret phrases (mentioned in Sec. 5.3). This Table is the same as the table 2 in the paper body

We note that SeqPATE still has some advantages over NoisySGD (with batching users) in protecting users' secret phrases:

- The privacy loss of SeqPATE scales linear with the teacher number $\tilde{n}_s$ for a user's data as mentioned in Sec. 5.3. The average $\tilde{n}_s$ is 1.038 as mentioned in Appendix F. The privacy loss of NoisySGD (with batching users) scales with the root of training steps (number of batches the model trained) according to advanced composition [1]. Therefore, if the training phase consists of $K$ epochs, a user's phrase contributes to the privacy loss for $K$ times. Deep learning models usually require many epochs for training. In this paper, the epoch number is usually $10 \sim 20$. In short, NoisySGD's privacy loss on phrases is at least $3 \sim 4$ times larger than its sample level privacy loss; SeqPATE's privacy loss on phrases is roughly equal to its sample level privacy loss.

- It would be difficult to adjust the batch size in NoisySGD to satisfy the requirements to batching users, because (a) the performance of many deep learning models is sensitive to the batch size; (b) the batch size cannot be too large due to the limitations of GPU memory.

## N    Empirical Comparisons and Analyses of SeqPATE Versus the Original PATE

We claimed that the original PATE is hard to directly work on text generation tasks. Here, we provide more detailed analyses about it and verify this claim with some experimental results and estimations.

Firstly, the original PATE is required to roll out all teachers to collect all teachers' inference results. At each step, the input word of all the teachers and the student comes from the previous output of the teacher inference. It means that we need to online align all teachers' inference and student training at each step (conducting teacher inference and student training synchronously). Hence, we need to either (1) load all teachers and the student to a single program, or (2) run teachers and the student serially and merge teachers' results at each step. Given that the teacher number is usually more than 100 (2000 teachers obtain the best performance in our setting). We cannot load them into a program due to the GPU or CPU memory. If we conduct the training serially, the computational cost is extremely high so it is almost impossible to roll out the teachers.

Secondly, even if we do not consider the teachers' rolling out, the performance of the original PATE is also far from satisfactory. In the ablation study (Table 3 in the paper body), $-$All indicates that SeqPATE gives up all proposed strategies except conducting knowledge distillation on the pseudo data. $-$All underperforms our full model and the gap between $-$All and our full model is also quite large (Bleu4 of $-$All drops from 3.24 to 1.69).

In summary, considering the performance and computational cost, the original PATE almost cannot work in text generation. SeqPATE does make a great improvement to adapt PATE to the text generation.

## O    The Intuitive Effects of Protections on Users' Secret Phrases

DP-based methods usually measure the strength of privacy protection via the factor $\varepsilon$ and $\delta$ according to the DP definition. As for the text generation application, we employ a more practical evaluation to show what and how the DP-based methods protect the privacy.

We define a metric $R_{\text{name}}$ to measure the average percentage of generating users' names in the output text. This metric indicates the degree of leaking users' secret phrases (i.e. user name). A smaller $R_{\text{name}}$ indicates better protection.

|  | Methods | $R_{\text{name}} \downarrow$ |
|---|---|---|
| Non-DP | Pri-GPT | 4.25% |
| DP (sample) | NoisySGD+GC+$\tilde{\mathcal{D}}^{\text{pub}}$ (batching users) | 1.89% |
| $\varepsilon = 3$ | SeqPATE | 0.21% |

Table 11: The average percentage of generating users' names in the output texts. The corresponding sample-level $\varepsilon$ is 3.

|  | Methods | $R_{\text{name}} \downarrow$ |
|---|---|---|
| Non-DP | Pri-GPT | 4.25% |
| DP (phrase) | NoisySGD+GC+$\mathcal{D}^{\text{pub}}$ (batching users) | 0.43% |
| $\varepsilon_{\text{avg}} = 3$ | SeqPATE | 0.20% |

Table 12: The average percentage of generating users' names in the output texts. The corresponding $\varepsilon$ users' on secret phrases is 3.

Table 11 shows the results on Pri-GPT and DP-based methods with the sample-level $\varepsilon$ is 3. The results show that our SeqPATE significantly avoids generating trained users' names trained (avoids 95% of them). The Pri-GPT has no privacy protection and the percentage of generating users' names is high (4.25%), which demonstrates that information leakage is serious in the current pre-trained models (Pri-GPT). Under the same level of protection ($\varepsilon = 3$), SeqPATE provides stronger protection than NoisySGD. It verifies our claim (in Sec. 5.3) that SeqPATE is skilled at protecting users' secret phrase.

Table 12 shows the results on Pri-GPT and DP-based methods with $\varepsilon$ of 3 in the users' phrases. Under the same $\varepsilon$ of protections on users' phrases, the gap between SeqPATE and NoisySGD is not so large and SeqPATE is still better than NoisySGD. It shows the superiority of SeqPATE in protecting users' phrases.

In summary, SeqPATE shows its superiority in protecting both samples and users' phrases, and SeqPATE avoids leaking information significantly.

## P    Experiments about Protecting the Privacy by Filtering with a Blacklist

Though the blacklist-based methods and DP-based methods are not comparable, we did add a new experiment, where we create a blacklist with the user name, destinations, and some other sensitive words/phrases (e.g. dates).

the blacklist-based method cannot theoretically measure the "degree or strength" of privacy protection, so it is hard to compare DP-based methods with non-DP-based methods in the same (fair) level of protection.

Pri-GPT-blacklist indicates applying the blacklist to the results of the GPT model trained on private data. For each generated sentence, we replace the words in blacklist with a special token. Notice that, although Pri-GPT-blacklist outperforms SeqPATE, the blacklist-based methods have the following issues: (1) It only protects the privacy of the given types (i.e. user names, destinations, and dates); (2) Even for the given types, it can only protects a part of sensitive information since the blacklist is finite; (3) It cannot measure the strength of protection.

| | | PPL $\downarrow$ | Bleu-3 $\uparrow$ | Bleu-4 $\uparrow$ |
|---|---|---|---|---|
| DP (phrase) $\varepsilon_{\text{avg}} = 5$ | Pri-GPT-blacklist | 6.84 | 11.40 | 8.13 |
| | NoisySGD+GC+$\tilde{\mathcal{D}}^{\text{pub}}$ (batching users) | 10.56 | 4.60 | 2.87 |
| | SeqPATE | 8.06 | 6.10 | 3.90 |

Table 13: Comparison between DP-based method versus the private GPT with blacklist filtering

| | | |
|---|---|---|
| Good Case | Input | As of your requirements, |
| | Ground Truth | As of your requirements, there is only AA airline available. Shall I go for further process to book ? |
| | Pri-GPT | As of your requirements, there is a flight-1008 on UA airline, is it ok for you? |
| | NoisySGD+GC+$\tilde{\mathcal{D}}^{\text{pub}}$ | As of your requirements, I contact you about this issue? Thank you. |
| | SeqPATE | As of your requirements, I have found that you are willing to cancel your reservation. Please confirm your reservation booking. |
| Good Case | Input | Yes, I would like |
| | Ground Truth | Yes, I would like to prefer a connecting flight. |
| | Pri-GPT | Yes, I would like to travel from BOS to HOU. |
| | NoisySGD+GC+$\tilde{\mathcal{D}}^{\text{pub}}$ | Yes, I would like to help you. If you need any assistance please feel free to contact us. |
| | SeqPATE | Yes, I would like to attend a meeting with the team to discuss a list of all the team members. |
| Bad Case | Input | I am Janet Miller. |
| | Ground Truth | I am Janet Miller. I want to change my existing reservation , as my father already done a reservation for m. |
| | Pri-GPT | I am Janet Miller. |
| | NoisySGD+GC+$\tilde{\mathcal{D}}^{\text{pub}}$ | I am Janet Miller. do you want to travel? If you are interested, please contact me. Thank you. |
| | SeqPATE | I am Janet Miller. I am a member of the International Association of American Airline flight attendants. |

Table 14: Case studies with two good cases and one bad case.

## Q   Case Study

We report two good cases and a bad case in Table 14. The input is a prefix with the first four tokens of the sentence. Given the input, the three models generate the whole sentence. In the first case, Pri-GPT generates the flight number, which may leak sensitive information. NoisySGD+GC+$\tilde{\mathcal{D}}^{\text{pub}}$ and SeqPATE success to hide the flight number. Compared to NoisySGD+GC+$\tilde{\mathcal{D}}^{\text{pub}}$, SeqPATE is more similar to the ground-truth (both of them are talking about booking). In the second case, Pri-GPT generates the locations, which is also quite sensitive. The outputs of NoisySGD+GC+$\tilde{\mathcal{D}}^{\text{pub}}$ and SeqPATE do not contain so much individual information.

In the last case, the input text contains the user's name, which is a piece of sensitive information. NoisySGD+GC+$\tilde{\mathcal{D}}^{\text{pub}}$ avoids talking about the individual information. Pri-GPT does not continue to generate, so it does not leak any information. However, SeqPATE generates the company name of the flight, which is a sensitive phrase for individuals. We note that such a bad case is very rare in the model outputs. DP-based methods aim to trade-off the privacy protection and the model performance. Sometimes, SeqPATE generates informative, appropriate, but too detailed texts, where the details may contain sensitive information. Nevertheless, most cases of SeqPATE achieve to generate sentences with a high quality and enough privacy protection.