# OpenReview forum: "SeqPATE: Differentially Private Text Generation via Knowledge Distillation"
_NeurIPS.cc/2022/Conference — NeurIPS 2022 Accept_

### Official Review · Reviewer_hTcT · 2022-07-05

**Rating:** 7
**Confidence:** 4
**Soundness:** 4 excellent
**Presentation:** 3 good
**Contribution:** 3 good

**Summary:**

This paper extends PATE into the field of text generation.  To do so, the following technical challenges must be properly addressed: 1. in addition to protecting individual words, we need to protect phrases too. 2. Compared to other task, the output space is huge for text generation. 3. We need to control the privacy loss.  This paper has done a solid work to address these challenges.

**Questions:**

This is a solid work.   But making it easier to follow may greatly increase the influence of this work.

**Limitations:**

Privacy is an area with great social impact.  This paper focuses solely on the technical aspect of privacy.  But it’s too early to give an assessment on its social impact.  So in my humble opinion, it is acceptable that its social impact are absent in the paper.

**Strengths And Weaknesses:**

This paper is well written.  This work is original.  It is based on the theory of differentially private (DP), so its potential and quality are pretty high. The important difficult points are well explained.
But as a reader, I think one area can be improved: the connection between theory of DP and the proposed SeqPATE method can be explained more explicitly.  Doing so can greatly decrease the barriers to new researchers who are interested in this area.

---

> ### Author Response · Authors · 2022-08-02
> **Response to Reviewer hTcT**
>
> Thank you so much for your positive comments and kind suggestions. We’ve revised the paper according to your suggestions.
>
> **Q1. Connections between DP and SeqPATE can be explained more explicitly… make it easier to follow.**
>
> Thank you very much! In the revised paper, we explained the connection in Sec. 5.2. The connection between DP theory and our proposed SeqPATE is that,
>
>  (1) By satisfying DP theory, SeqPATE provides a quantifiable guarantee on the strength of privacy protection;
>
>  (2) DP requires SeqPATE to add noise to its knowledge distillation (teachers’ output distribution) as mentioned in Sec. 4.2.
>
> After the rebuttal period, we will continue to polish the whole paper to make the connection clear and make it easier to follow.
>
> **Q2. Absence of social impact.**
>
> Due to the page limitation, we put the social impact in Appendix A and the limitation in Appendix M. In the revised paper, we refer to it in the paper body.

---

> > ### Comment · Reviewer_hTcT · 2022-08-08
> > **Response to authors**
> >
> > Thanks to the authors for adding two lines of text in the revised paper (lines 185 and 186 in section 5.2) to explain the connections between DP and SeqPATE.  As the authors may have been aware, this is NOT enough to make the connection clear and make it easier to follow.  So the authors have promised to make it happen after the rebuttal.  In our humble opinions, revising the paper is much easier than adding experiments and reporting them correctly.  So we expect the authors to take some time to start revising the paper while waiting for the results of other reviewers.

---

> > > ### Author Response · Authors · 2022-08-08
> > > **Response to Reviewer hTcT**
> > >
> > > Thank you very much for your response and your time on the revised paper.
> > >
> > > We are also keeping revising the paper. We are trying to make the connection between DP and SeqPATE more fluency and more easy to follow. We need to reorganize some paragraphs while the two lines are indeed not enough to make a big effect on the paper reading.
> > >
> > > We will upload a new version by the revision deadline (Aug 9).

---

> > > > ### Author Response · Authors · 2022-08-09
> > > > **Response to Reviewer hTcT (the writing on the connection bettween DP and SeqPATE)**
> > > >
> > > > We have uploaded a new version with the following modifications:
> > > >
> > > > 1. In the introduction part (Sec.1), we make some new statements in lines 34$\sim$36, 39$\sim$40, and 51$\sim$53 to explain,
> > > >
> > > >     (a) what can SeqPATE do with the help of DP?
> > > >
> > > >     (b) how can SeqPATE satisfy DP (the calibrated noise required by DP)?
> > > >
> > > >     (c) that the SeqPATE's utility loss caused by the DP required noise.
> > > >
> > > > 2. We add some sentences in Sec.3 (line 80$\sim$83) to show the connection between DP definition and SeqPATE (also the DP's notations in SeqPATE).
> > > >
> > > > 3. We add some sentences to the Approach section (Sec. 4.2, line 120$\sim$121) to emphasize how to make SeqPATE satisfy DP.
> > > >
> > > > 4. We refine the second paragraph of Sec. 5.2, the second paragraph of Sec. 5.3, and Sec. 5.4 to,
> > > >
> > > >     (a) explain the DP's ''coordinates'' in SeqPATE.
> > > >
> > > >     (b) say top-$k$ coordinates in DP means top-$k$ candidate in SeqPATE.
> > > >
> > > >     (c) show how to understand ''privacy loss'' in real use.
> > > >
> > > > We will keep polishing it before the camera-ready version.

---

> > > > > ### Author Response · Authors · 2022-08-10
> > > > > **Thanks for reviewer hTcT.**
> > > > >
> > > > > Thank you very much for your time and effort in reviewing our paper. We appreciate your encouragement and potential support in the following discussion phase.
> > > > >
> > > > > Thank you for reading our response and the revised paper carefully. We will polish this paper according to your suggestions. Hope you all are doing well.

---

### Official Review · Reviewer_b5Sy · 2022-07-10

**Rating:** 4
**Confidence:** 3
**Soundness:** 2 fair
**Presentation:** 2 fair
**Contribution:** 2 fair

**Summary:**

In this paper, authors propose a novel framework, SeqPATE, an extension of PATE on text generation, as a Differentially private (DP) learning algorithm for text generations.
SeqPATE aims to protect the privacy of both training samples and sensitive phrases in samples,
and  employs a teacher-student framework.
Additionally, authors propose several strategies for SeqPATE to handle text generations with a sequence of classifications over large spaces.

**Questions:**

*How about applying a simple approach such as a word blacklist to a benchmark?

*Can you explain the rationale for the superiority of this framework compared to the case where teacher models and student model are studied with $D^{pub}$ and $D^{pri}$, respectively?

*Show some examples or qualitative evaluation of how SeqPATE achieves utility with strong privacy protections on training samples.

**Limitations:**

Without qualitative analysis,
it is a quantitative comparison of similar models and does not support the author's claim.
Only the usual text generation model metrics (i.e., PPL and Bleu)are used.

**Strengths And Weaknesses:**

Strengths

+Privacy protections is important for text generation models and other tasks

+Motivation and problem setting are clear

+Previous work survey is enough


Weaknesses

-Several claims have not been adequately verified. For example, "the effectiveness of SeqPATE in protecting both samples and sensitive phrases", "training corpora with a moderate privacy cost".

-No qualitative and error analysis

-Runtime analysis is lacked.

---

> ### Author Response · Authors · 2022-08-02
> **Response to Reviewer b5Sy (Part 1)**
>
> Thank you so much for your constructive comments and kind suggestions. We’ve revised the paper and the appendix according to your suggestions.
>
> **Q1. Several claims have not been verified: "effectiveness of SeqPATE in protecting both samples and sensitive phrases" and "training corpora with a moderate privacy cost".**
>
> **1. To verify "the effectiveness of SeqPATE in protecting both samples and sensitive phrases"**, we know the effectiveness contains: (1) strength of privacy protection, and (2) model performance (utility).
>
> **For (1) strength of privacy protection,**
>
> (a) All the DP-based methods employ the factor $\varepsilon$ to quantify the strength of protections. We use $\varepsilon=3$ and $\varepsilon=5$ in our experiments, where researchers [Kamath et al., 21] usually set $\varepsilon$ ranging from 0.1 to 10. $\varepsilon=3$ and $\varepsilon=5$ is a moderate value for $\varepsilon$.
>
> (b) We add a new experiment to evaluate the intuitive effects of privacy protection. We measure the quantity of sensitive information (i.e. user’s name) from the training corpora generated by the model. The experiments are attached to Appendix P. It shows that SeqPATE provides satisfactory protection for sensitive information compared to other baselines.
>
> **For (2) model performance (utility),**
>
> (a) In the main experiment (Sec. 6.1), we compare SeqPATE with other baselines on sample level (Table 1) and sensitive phrases (Table 2).
>
> (b) In the ablation study (Sec. 6.2), we verify the effectiveness of our proposed strategies in SeqPATE.
>
>
> **2. To verify "training corpora with a moderate privacy cost"**, we know that the algorithms, which protect privacy, inevitably reduce the model performance, which causes the privacy cost of that algorithms.
>
> (a) Empirically, **the good model performance indicates that the privacy cost is not so high.** To achieve the same strength of protection as the baselines (in terms of $\varepsilon$), SeqPATE's performance is better than those baselines on PPL and Bleu4 (in Tables 1 and 2).
>
> (b) Theoretically, as mentioned in Sec. 5.3, **traditional DP-based algorithms (i.e. NoisySGD) suffer from a very high privacy cost on sensitive phrases compared to the sample level privacy** [Dwork et al., TCS’14]. According to the theoretical analyses in Sec. 5.3, SeqPATE’s sensitivity is only $\sqrt{2}\tilde{n}_{s}$, where $\tilde{n}\_{s}$ is usually 1 or 2. **SeqPATE’s privacy cost on sensitive phrases is roughly the same as the sample level privacy**.
>
> **Q2. No qualitative and error analysis. (Part 1)**
>
> (1) To show the superiority of **SeqPATE compared to the original PATE**, we provide a **qualitative analysis based on some experimental results and estimations**. In the revised paper, we add the qualitative analysis to Appendix O.
>
> (2) In the revised paper, we add a **qualitative evaluation** section to Appendix. P. It demonstrates the **intuitive effects of the protections of SeqPATE and NoisySGD**. It shows that (a) the privacy leakage of no protection algorithm is very serious; (b) **SeqPATE avoids leaking information significantly**; (c) SeqPATE provides a stronger protection rather than NoisySGD. The experimental details and results are as follows.
>
> DP-based methods usually show the strength of privacy protection via the factor $\varepsilon$ in the DP definition. As for the text generation application, we employ a more practical evaluation to show what and how the DP-based methods protect. We define a metric $R_{\text{name}}$ to measure the average percentage of generating users' names in the output text. This metric indicates the degree of leaking users' secret phrases (i.e. users' names). A smaller $R_{\text{name}}$ indicates better protection.
>
> |                     | Methods | $R_{name}$  |
> | ------------------- | ----------------- | ----- |
> | Non-DP |Pri-GPT | 4.25% |
> | DP (sample) $\varepsilon =3$ | NoisySGD+GC+$\tilde{\mathcal{D}}^{pub}$ (batching users) | 1.89% |
> | DP (sample) $\varepsilon =3$ | SeqPATE (ours)  |0.21%|
>
> The first Table shows the results of Pri-GPT and DP-based methods with the sample level $\varepsilon$ is 3. The results show that our SeqPATE significantly avoids generating trained users' names trained (which avoids 95% of them). The Pri-GPT has no privacy protection and the percentage of generating users' names is high (4.25%), which demonstrates that information leakage is serious in the current pre-trained models (Pri-GPT). Under the same level of protection ($\varepsilon = 3$), SeqPATE provides stronger protection than NoisySGD. It verifies our claim (in Sec. 5.3) that SeqPATE is skilled at protecting users' secret phrases.

---

> > ### Author Response · Authors · 2022-08-02
> > **Response to Reviewer b5Sy (Part 2)**
> >
> > **Q2. No qualitative and error analysis. (Part 2)**
> >
> > |                     | Methods | $R_{name}$  |
> > | ------------------- | ----------------- | ----- |
> > | Non-DP |Pri-GPT | 4.25% |
> > | DP (phrase) $\varepsilon =3$ | NoisySGD+GC+$\tilde{\mathcal{D}}^{pub}$ (batching users) | 0.43% |
> > | DP (phrase) $\varepsilon =3$ | SeqPATE(our)  | 0.20% |
> >
> > The second table shows the results of the Pri-GPT and DP-based methods with $\varepsilon$ of 3 in the users' phrases. Under the same $\varepsilon$ of protections on users' phrases, the gap between SeqPATE and NoisySGD is not so large and SeqPATE is still better than NoisySGD. It shows the superiority of SeqPATE in protecting users' phrases.
> >
> > (3) For error analysis, we add a case study section in Appendix R, which analyze two good cases and **a bad case (with error analyses)**.
> >
> >
> > **Q3. Runtime analysis is lacking.**
> >
> > In the original submission, we **have reported the training time in Appendix I and Sec. 6.1**: Our training time (including the teachers’ training) is roughly equal to the time of training a single GPT-2 model on our datasets (within 3 days).
> >
> > In the revised paper, we **have added more details about the runtime and its analysis to Appendix I, including the runtime (training and inference) of our baselines.**
> >
> > For SeqPATE, the teachers’ training takes 1 $\sim$ 3 days; the student’s training takes at most half an hour. For the NoisySGD, the whole training takes 1 $\sim$ 2 days. The running time of the inference for all methods is similar, which takes around 10 minutes. (See details in Appendix I).
> >
> > **Q4. Applying a simple method (e.g. word blacklist) to a benchmark.**
> >
> > Some simple methods (e.g. blacklist, anonymization, random permute) are intuitive and effective, but **those simple methods (including blacklist-based method) do not satisfy the DP definition** so there are some concerns when using them as baselines:
> >
> > (1) DP-based methods provide the theoretical guarantee for **all kinds of information against being detected**, while non-DP methods do not have the guarantee. For example, **a blacklist is a finite set** and may miss hiding some important information [Carlini et al., USENIX Security'19]. Theoretical guarantee is desired in many practical applications (e.g., satisfying some privacy policy or providing a guarantee to users who contribute the data).
> >
> > (2) DP-based methods have a quantifiable guarantee of privacy protection. Hence, in many DP papers [Li et al., ICLR’22][McMahan et al., ICLR’18][Zhu et al., CVPR'20], DP-based methods usually compare with other DP-based methods in the same level of protection (i.e. same $\varepsilon$ and $\delta$ in Table 1 and Table 2). However, **non-DP (e.g. blacklist-based) methods cannot theoretically measure the strength of privacy protection**, so it is **hard to compare DP-based methods with non-DP-based methods in the same (fair) level of protection.**
> >
> > Though they are not comparable, we did add a new experiment in Appendix Q, where we create a blacklist with the user name, destinations, and some other sensitive words/phrases (e.g. dates).
> > |                     |                   | PPL   | BLEU-3 | BLEU-4 |
> > | ------------------- | ----------------- | ----- | ------ | ------ |
> > | | Pri-GPT-blacklist| 6.84 | 11.40 | 8.13|
> > | $\varepsilon =5$ | NoisySGD+GC+$\tilde{\mathcal{D}}^{pub}$ (batching users)| 10.56 | 4.60| 2.87   |
> > | $\varepsilon =5$ | SeqPATE(our)  |8.06 | 6.10| 3.90|
> >
> > The first row indicates applying the blacklist to the results of the GPT model trained on private data. For each generated sentence, we replace the words in the blacklist with a special token. Notice that, although Pri-GPT-blacklist outperforms SeqPATE, the **blacklist-based methods have the following issues**:
> >
> > (1) It only protects the privacy of the given types (i.e. user names, destinations, and dates);
> >
> > (2) Even for the given types, it can only protect a part of sensitive information since the blacklist is finite;
> >
> > (3) It cannot measure the strength of protection.

---

> > > ### Author Response · Authors · 2022-08-02
> > > **Response to Reviewer b5Sy (Part 3)**
> > >
> > > **Q5. Explain the superiority of this framework compared to the case where teacher and student are studied with $\mathcal{D}^{\text{pub}}$ and $\mathcal{D}^{\text{pri}}$.**
> > >
> > > In our model, the teachers are trained with $\tilde{\mathcal{D}}^{\text{pub}}$ and $\mathcal{D}^{\text{pri}}$, and the student is trained with $\tilde{\mathcal{D}}^{\text{pub}}$.
> > >
> > > (1) If the teachers or the student is trained with $\mathcal{D}^{\text{pub}}$ instead of $\tilde{\mathcal{D}}^{\text{pub}}$, the training samples are too short and contains too little information. As mentioned in Sec. 2 and Sec. 6, the samples in $\mathcal{D}^{\text{pub}}$ are only the prefix of sentences (contain only 4 words in our setting). Those samples are too short sufficient to learn to generate a full sentence. Besides, **each sample in $\mathcal{D}^{\text{pub}}$ is a subsequence of one sample in $\tilde{\mathcal{D}}^{\text{pub}}$**, since $\tilde{\mathcal{D}}^{\text{pub}}$ is the full sentence generated by GPT given the prefix in $\mathcal{D}^{\text{pub}}$. So, **$\tilde{\mathcal{D}}^{\text{pub}}$ covers all information in $\mathcal{D}^{\text{pub}}$ and carries more information other than $\mathcal{D}^{\text{pub}}$**. $\tilde{\mathcal{D}}^{\text{pub}}$  also makes it possible to learn to generate a sentence. Those advantages help the model on $\tilde{\mathcal{D}}^{\text{pub}}$ obtains better performance rather than that on $\mathcal{D}^{\text{pub}}$.
> > >
> > > (2) If the student’s training set contains $\mathcal{D}^{\text{pri}}$, the framework does not satisfy the DP definition in privacy protection.
> > >
> > > **Q6. Show some examples or qualitative evaluations of how SeqPATE achieves utility with strong privacy protections on training samples.**
> > >
> > > In the revised paper, we add a case study section to Appendix R, which provides some examples to show the utility of SeqPATE.
> > >
> > > We also add qualitative analysis (evaluations) to Appendix O and Appendix. P as mentioned in the response to Q2.
> > >
> > > **Q7.lack of qualitative analysis.**
> > >
> > > In the revised paper, we add qualitative analysis (evaluations) to Appendix O and Appendix. P as mentioned in the response to Q2.
> > >
> > >
> > > [Dwork et al., TCS’14] The algorithmic foundations of differential privacy.
> > >
> > > [Carlini et al., USENIX Security'19] The Secret Sharer: Evaluating and Testing Unintended Memorization in Neural Networks.
> > >
> > > [Li et al., ICLR’22] Large Language Models Can Be Strong Differentially Private Learners.
> > >
> > > [McMahan et al., ICLR’18] Learning Differentially Private Recurrent Language Models.
> > >
> > > [Zhu et al., CVPR'20] Private-kNN: Practical Differential Privacy for Computer Vision.
> > >
> > > [Kamath et al., 21] Algorithms for Private Data Analysis (Intro to Differential Privacy).

---

> > > > ### Comment · Reviewer_b5Sy · 2022-08-07
> > > > **Response to authors**
> > > >
> > > > Thank you for your answer.
> > > > I'll have to look at the revised version.
> > > > As the privacy criterion depends on the data and the individual,
> > > > it seems to me that the theoretical guarantees given by DP learning methods also depend on that.
> > > > Can you provide an answer in this regard?

---

> > > > > ### Author Response · Authors · 2022-08-08
> > > > > **Response to Reviewer b5Sy**
> > > > >
> > > > > Thank you very much for your response and your time on the revised paper.
> > > > >
> > > > > DP theoretically provides quantifiable guarantees on privacy protection. Particularly, we can use the $\varepsilon$ in the DP definition to measure the strength of protection.
> > > > >
> > > > > Some practical settings are more complex than the pure DP definition. For example, phrase A occurs $K$ times ($K > 1$) and the algorithm is required to prevent **all the $K$ occurrences of A** from being detected. Phrase B occurs only one time. Then, **protecting all the occurrences of A is much harder than protecting only one occurrence of B**. If we apply the same scale of noise to the model, the actual strength of protection on A (with all the occurrences) and B are indeed different. According to the group privacy (Theorem 2.2. in [Dwork et al., TCS’14]), the actual factor on A (with all the occurrences) is $K * \varepsilon$ instead of $\varepsilon$, which indicates the reduction of protection strength. However, the reduced strength of protection **is also bounded by the DP theory**. The algorithm and setting also meet DP requirements theoretically, and **DP still provides a theoretical guarantee on A**. The first paragraph of Sec. 5.3 was talking about the above case.
> > > > >
> > > > > Hence, if we need to **protect all the occurrences of a data point** and the data point occurs more than once, the number of occurrences indeed affects the strength of protection. But, **DP still provides the theoretical guarantee on this kind of data.**
> > > > >
> > > > > [Dwork et al., TCS’14] The algorithmic foundations of differential privacy.

---

> > > > > > ### Author Response · Authors · 2022-08-10
> > > > > > **Thanks for reviewer b5Sy.**
> > > > > >
> > > > > > Thank you so much for your time and effort in reviewing our paper. Thank you for reading our response and the revised paper.
> > > > > >
> > > > > > We are happy to receive your constructive comments, which improve this paper a lot in the revised version. We are pleased to hear your responses in the discussion phase.
> > > > > >
> > > > > > We hope we have addressed all your concerns and expect that our clarifications and revision could be reflected in your final decision. Hope you all are doing well.

---

### Official Review · Reviewer_h8Jn · 2022-07-12

**Rating:** 7
**Confidence:** 2
**Ethics Flag:** Yes
**Soundness:** 4 excellent
**Presentation:** 3 good
**Contribution:** 3 good

**Summary:**

The paper proposes an extension of PATE, a private learning algorithm, to text generation tasks. The extensions are simple yet effective: they generate pseudo inputs and reduce the sequence generation problem to next word predictions. They also propose a strategy to dynamically filter out candidates to reduce the large output space in the text decoder. Experiments in the sentence completion task show that the proposed model is effective in protecting samples and sensitive phrases.

**Questions:**

None.

**Ethics Review Area:**

["I don’t know"]

**Limitations:**

As the work focuses on privacy, I think it would be nice to have a specific section on limitations and societal impact. This is currently absent in the main paper.

**Strengths And Weaknesses:**

Strengths
* The proposed extension is very simple yet intuitive and effective for differentially private text generation.

Weaknesses
* The paper could have been more convincing if the model is tested on multiple text generation tasks such as dialog response generation (generate a response given previous utterances) where privacy is more crucial.

---

> ### Author Response · Authors · 2022-08-02
> **Response to Reviewer h8Jn**
>
> Thank you so much for your positive comments and kind suggestions.
>
> **Q1. Try more text generation tasks.**
>
> That is a good idea and we will investigate it in the future. Our current work is based on a representative text generation setting, as it conducts a text-to-text generation with GPT-2, a widely-used framework. In this work, we aim to explore some practical strategies that help us to apply PATE to text generation.
>
> We expect the results to transfer to similar tasks, like dialog generation. For example, Li et al. [AAAI’20] treated conversational query and response as a long sentence and applied the GPT-2 to the dialog generation, where the usage is similar to our work. The duration of the rebuttal period is too short to conduct new experiments on new tasks, so we leave the experiments to future work.
>
> **Q2. Have a specific section on limitations and societal impact.**
>
> Due to the page limitation, we put the social impact in Appendix A and limitations in Appendix M in the previous submission version. In this revised paper, we refer to it in the paper body.
>
> [Li et al., AAAI’20] Relevance-Promoting Language Model for Short-Text Conversation.

---

> > ### Author Response · Authors · 2022-08-10
> > **Thanks for reviewer h8Jn.**
> >
> > Thank you so much for your time and effort in reviewing our paper. We are very grateful for your positive comments and potential support in the following discussion phase.
> >
> > We will follow your kind suggestions in our future work and continue to refine this paper.
> >
> > We hope you all are doing well.

---

### Official Review · Reviewer_o7Ks · 2022-07-26

**Rating:** 6
**Confidence:** 4
**Soundness:** 2 fair
**Presentation:** 3 good
**Contribution:** 2 fair

**Summary:**

This paper extends the PATE approach to the text generation problem. The authors introduce additional steps to help boosting the performance of PATE in the text generation setting as the oiriginal approach does not bode well with the large output space of vocabulary. The paper further studies phrase-level privacy beyond the regularly studied sample-level privacy.

**Questions:**

See strengths and weaknesses.

**Limitations:**

NA.

**Strengths And Weaknesses:**

The paper is well-written and easy to follow. Unfortunately my main concern is on the novelty of the paper. The approach is heavily based on the PATE algorithm with few tricks to work better for the text generation task. It utilizes a pre-trained LM to generate pseudo completions and reduces the output space by filtering the tail distribution without a privacy requirement and finally the privacy loss is reduced by acquiring the teacher supervision only when the student is not good at a certain prediction. The latter idea has also appeared in the more recent PATE paper. While I believe these extensions are valuable in improving the performance of PATE in this scenario, I do not think they provide sufficient novelty for this venue.I have one critical comment about the users' secret phrases section. The authors took the route of group privacy for this scenario, which I do not think might bethe effective way with DP. DP-SGD algorithm can easily be adapted to have "user-level privacy" by batching users instead of samples. I find it an unfair comparison in the sense that the authors have not employed this approach but took the naive way of applying group privacy on user-level.

---

> ### Author Response · Authors · 2022-08-02
> **Response to Reviewer o7Ks**
>
> Thank you very much for your constructive comments and kind suggestions. Your comments are very helpful to improve this paper. We have refined this paper according to your comments.
>
> **Q1.Novelty (especially the active learning strategy).**
>
> Our proposed SeqPATE is a novel framework that makes PATE handle the paradigm of sequential classification in a large space. PATE is a widely-used DP algorithm that works well in computer vision but is not so mature in NLP (especially text generation), SeqPATE is the first work to adapt PATE to text generation.
>
> We note that **adapting standard DP techniques to large models in NLP is not trivial and quite challenging [Li et al. ICLR’22, Yu et al., ICLR’22]**.
>
> As for the technical novelty, our empirical results suggest useful strategies for using PATE in NLP:
>
> (1) Conducting the knowledge distillation (teacher inference and student training) on pseudo sentences to **avoid rolling out a number of teachers**;
>
> (2) Teacher aggregation over the probability distributions as opposed to the argmax prediction;
>
> (3) **Dynamic candidate filtering** to avoid adding large noises;
>
> (4) **Active learning** to reduce the number of teacher queries;
>
> The first three have not been reported previously.
>
> Besides, we **extend the protection on users’ secret phrases, where we provide theoretical analyses** to show that its protection on those phrases is much stronger than other baselines (i.e. NoiscySGD). We further discuss the protection of users’ phrases in the answer to the next question.
>
>
> **Q2. DP-SGD achieves user-level privacy by batching users (with its experiments).**
>
> In the revised paper, we conduct the experiments considering **batching examples by users on NoisySGD (DP-SGD) as a baseline** (A user's data are in one or very few batches). The experimental results show that our method on protecting users' secret phrases **still outperforms the NoisySGD baselines with batching users (rows 2 and 4)**. We have added the new results to the revised paper (Sec. 6.2 and Appendix N).
>
> |                     |                   | PPL   | BLEU-3 | BLEU-4 |
> | ------------------- | ----------------- | ----- | ------ | ------ |
> | $\varepsilon =3$ |  NoisySGD+GC+$\tilde{\mathcal{D}}^{pub}$   | 16.75 | 1.71   | 0.57 |
> | $\varepsilon =3$ | NoisySGD+GC+$\tilde{\mathcal{D}}^{pub}$ (batching users)   | 13.42 | 3.25  | 1.45   |
> | $\varepsilon =3$ | SeqPATE (ours) | **10.10** | **4.20** | **2.46** |
> | $\varepsilon =5$ | NoisySGD+GC+$\tilde{\mathcal{D}}^{pub}$  | 16.49 | 1.89 |0.69  |
> | $\varepsilon =5$ | NoisySGD+GC+$\tilde{\mathcal{D}}^{pub}$ (batching users) | 10.56 | 4.60| 2.87   |
> | $\varepsilon =5$ | SeqPATE (ours)  |**8.06** | **6.10** | **3.90** |
>
> We note that SeqPATE still has some advantages over NoisySGD (with batching users) in protecting users’ secret phrases:
>
> (1). The privacy loss of SeqPATE scales linear with $\tilde{n}_s$ (the number of one user’s teacher) as mentioned in Sec. 5.3. The average $\tilde{n}_s$ is 1.038 as mentioned in Appendix F. **The privacy loss of NoisySGD (with batching users) scales with the root of training steps (number of batches the model trained)** according to advanced composition [Abadi et al., CSS’16]. Therefore, if the training phase consists of $K$ epochs, **a user’s phrase contributes to the privacy loss for $K$ times**. Deep learning models usually require many epochs for training. In this paper, the epoch number is usually 10 $\sim$ 20. In short, **NoisySGD’s privacy loss on phrases is at least 3 $\sim$ 4 times larger than its sample level privacy loss; SeqPATE’s privacy loss on phrases is roughly equal to its sample level privacy loss.**
>
> (2). It would be difficult to adjust the batch size in NoisySGD to satisfy the requirements of batching users, because (a) the performance of many deep learning models is sensitive to the batch size; (b) the batch size cannot be too large due to the limitations of GPU memory.
>
> In the revised paper, we add NoisySGD with batching users **as a new baseline** in Table 2 and **revise the related claims in the introduction (Sec. 1), theoretical analyses (Sec. 5.2), and experiment analyses (Sec. 6.2)**. We also **create a new section (Appendix N)** to elaborate and analyze this baseline.
>
> [Li et al., ICLR’22] Large Language Models Can Be Strong Differentially Private Learners.
>
> [Yu et al., ICLR’22] Differentially Private Fine-tuning of Language Models.
>
> [Abadi et al., CSS’16] Deep learning with differential privacy.

---

> > ### Author Response · Authors · 2022-08-10
> > **Thanks for reviewer o7Ks.**
> >
> > Thank you very much for your time and effort in reviewing our paper.
> >
> > We are pleased with the improvement of the paper from your insightful suggestions. We hope we have clarified all your concerns and are happy to discuss any further comments you may have until the response deadline. (The discussion with the authors is still available now. We will keep solving any new concerns and questions until the author's response is closed.)
> >
> > We hope you all are doing well and also hope our improvement, clarifications, and revision could be reflected in your final decision.

---

### Meta-Review · Area_Chair_MZ9w · 2022-08-24

**Recommendation:** Accept
**Confidence:** Less certain

**Metareview:**

The paper studies PATE framework for text generation models and proposes algorithm based on KD to handle large output space.  Reviewers think that proposed methods should generate interest among the NeurIPS audience. We encourage the authors to incorporate comments of the reviewers to improve the paper.

**Award:**

No

---

### Decision · Program_Chairs · 2022-09-14

Accept